# Dynamic pathogen detection and social feedback shape collective hygiene in ants

Barbara Casillas-Pérez[1,3], Katarína Boďová[2,3], Anna V. Grasse[1], Gašper Tkačik ®[1] ✉ & Sylvia Cremer ®[1] ✉

Cooperative disease defense emerges as group-level collective behavior, yet how group members make the underlying individual decisions is poorly understood. Using garden ants and fungal pathogens as an experimental model, we derive the rules governing individual ant grooming choices and show how they produce colony-level hygiene. Time-resolved behavioral analysis, pathogen quantification, and probabilistic modeling reveal that ants increase grooming and preferentially target highly-infectious individuals when perceiving high pathogen load, but transiently suppress grooming after having been groomed by nestmates. Ants thus react to both, the infectivity of others and the social feedback they receive on their own contagiousness. While inferred solely from momentary ant decisions, these behavioral rules quantitatively predict hour-long experimental dynamics, and synergistically combine into efficient colony-wide pathogen removal. Our analyses show that noisy individual decisions based on only local, incomplete, yet dynamically-updated information on pathogen threat and social feedback can lead to potent collective disease defense.

Collective action is often more successful than solitary action. This is because simple individual decisions based on local, noisy, or incomplete information can interact through feedback to generate complex and efficient collective behavior, be it in systems of interacting genes[1], neurons[2,3], cells in a tissue[4], among single-celled organisms[5], or in animal collectives[6–11]. Social insects like the social bees and wasps, the ants and termites – where selection acts on performance at the colony level – are a paradigmatic example for the emergence of self-organized collective behavior[12]. Their colonies master collective transport, food collection, as well as nest choice and architecture[13–19]. Moreover, social insects perform cooperative disease defense that gives rise to colony-level protection, or social immunity[20]. Not only do social insects decrease disease transmission by modulating their social interaction networks[21,22], they also reduce overall disease risk by active sanitary care behaviors like grooming-off infectious particles from exposed colony members[23–26]. While the hygienic repertoire of social insects has been described previously[20,27–31], the quantitative decision rules

underlying these behaviors and how they combine into colony-wide disease protection remain largely unexplored.

To understand the individual decision-making process that forms the basis of emergent collective hygiene, we here put ants into an experimental situation where they could choose how to distribute their sanitary care between two group members carrying different loads of an infectious fungal pathogen. Observation of all individual and mutual grooming events in a time-resolved manner and quantification of the spore load of each ant after the end of the experiment allowed us to infer – for every time window during the experiment – the spore load of both treated ants. Based on this information we could determine, for each grooming event performed, whether the groomed ant was the one with the higher or lower current spore load, which revealed that ants over-proportionally groom the individual with the higher current – yet not necessarily higher initial – spore load. We complemented our experimental work by probabilistic modeling[32] to infer universal decision rules and factors in the ant's recent experience

[1]ISTA (Institute of Science and Technology Austria), Am Campus 1, AT-3400 Klosterneuburg, Austria. [2]Department of Mathematical Analysis and Numerics, Faculty of Mathematics, Physics, and Informatics, Comenius University, Mlynska Dolina SK-84248 Bratislava, Slovakia. [3]These authors contributed equally: Barbara Casillas-Pérez, Katarína Boďová. ✉e-mail: gasper.tkacik@ist.ac.at; sylvia.cremer@ist.ac.at

that determine their individual grooming decisions. This joint approach shows that an ant's grooming activity depends on the perceived pathogen threat emerging from its colony members, but also on the ant's sensitivity to social cues provided by the colony members about its own infectivity. In particular, an individual ant grooms other ants more, the more spores it perceived on others in the recent past, yet it engages less in grooming after being groomed itself. As the frequency of being groomed is driven by an ant's own spore load, this suppressive effect is hence strongest in individuals that carry highest spore load, so that social feedback effectively prevents the most infectious colony members from caregiving. Modeling further revealed that the observed grooming decisions of individual ants can be best explained by a spore-load dependent, yet probabilistic, rule of whom to groom during sequential encounters with its colony members. Such a simple local rule does not require global knowledge about the spore load of all colony members, thus likely allowing efficient pathogen removal also in large colonies. We further give experimental evidence that preventing the ants from making free grooming decisions leads to reduced group-level pathogen removal efficiency, revealing the importance of informed individual choices for colony-level hygiene. Together, this shows that simple individual rules integrating both threat level and social feedback assure that the most

infectious colony members receive most but engage least in caregiving, leading to highly efficient social immunity[20] and reduced risk of disease spread[33] at the colony level.

## Results

### Exposing ants to a choice situation for sanitary care

To derive individual decision rules for sanitary care in ants, we set up different experimental combinations of individual- and group-level pathogen loads, comprehensively observed the behaviors of individual group members at 1/15 sec-temporal resolution, and quantified pathogen removal and transmission. We restricted our study to worker ants as queens do not engage in social immunity measures[34,35]. Since collective phenomena in ants emerge already in small group sizes of six individuals[36], we formed groups ($n = 99$) of six workers of the ant *Lasius neglectus*, to observe all individual and pairwise sanitary actions. We treated two of the workers with either a high (F) or a low (f) dose of the fungal pathogen *Metarhizium robertsii*, or a non-pathogenic control (C), and combined them to create groups differing both in overall pathogen load and the load differences between the two treated individuals. The untreated four nestmates therefore either faced a clear (FC, fC), less distinct (Ff) or no (FF, ff, CC) initial spore load difference between the two treated individuals (Fig. 1a; Supplementary

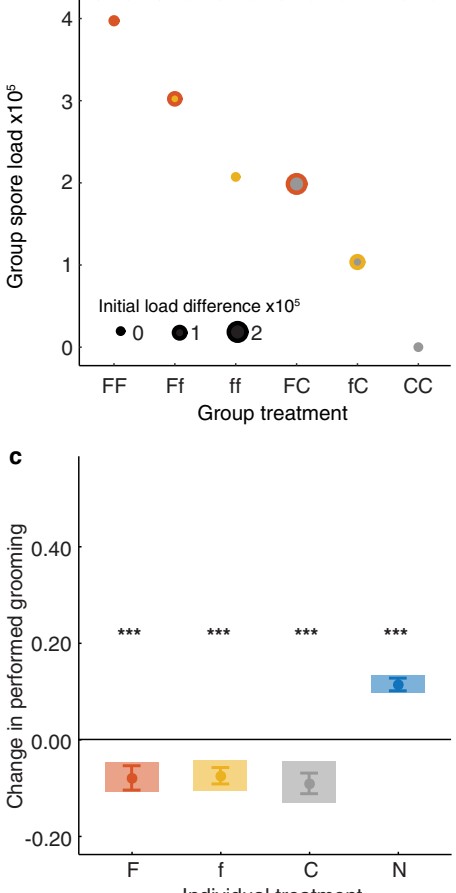

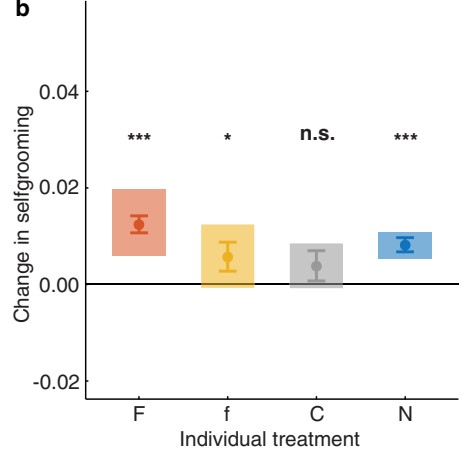

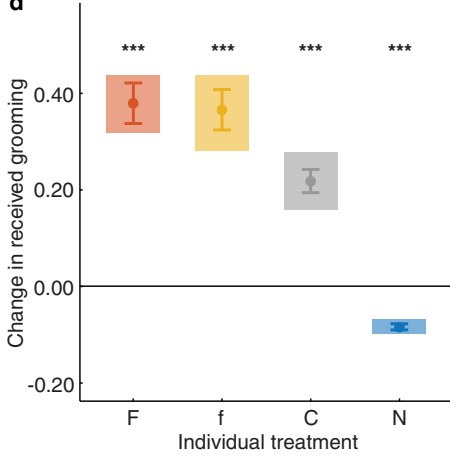

**Fig. 1 | Individual and collective hygiene are triggered by pathogen exposure.** **a** Experimental setup of six treatment groups, each consisting of four untreated ants (nestmates) and two ants treated with varying pathogen load (red F, high; yellow f, low; gray C, control). Group spore load based on the exposure loads applied to F- and f-treated ants (as determined directly after exposure for $n = 30$ individuals each) and initial spore load difference between the two treated ants are shown per treatment group (medians depicted, see "Methods" section for interquartile ranges). Treatment-induced behavioral changes, reported as difference from the pre-treatment period (zero line) in fraction of effective time spent in **b** selfgrooming their body (resp. head, Supplementary Table 2), **c** performed and **d** received allogrooming, for treated ants and their untreated nestmates (blue N). Mean ± sem depicted in opaque colors, shades show 95% CI, $n = 594$ ants, 99 replicates, Supplementary Table 1). Two-sided $p$-values adjusted for multiple testing of paired Wilcoxon tests post- vs pre-treatment depicted by ***$p \leq 0.001$ (details given in Supplementary Table 2), *$p = 0.023$, n.s. $p = 0.105$. Source data are provided as a Source Data file.

Table 1; $n$ = 16-17 replicates per group treatment). Individual ants could be distinguished by a unique color-code applied on their body (yet with color not revealing ant treatment to the observer), and the spores of pathogen-treated ants by distinct genetically-encoded labels (fluorescent GFP vs. RFP; Supplementary Fig. 1). We continually quantified the sanitary behaviors of each ant for 30 min before and 90 min after treatment, in particular, self-hygiene (selfgrooming) and grooming others (allogrooming). Grooming is a common sanitary behavior in social insects, combining the removal of infectious particles from the body surface by the insects' mouth parts, subsequent compacting and disinfection in specific infrabuccal pockets in the head, and later expulsion of inactivated pathogen as pellets[23,24,37]. After the end of the experiment, we sampled each ant and determined the number and origin of spores separately for its head and body, as well as for the pellets produced per group, using sensitive quantitative PCR (droplet-digital PCR targeting the gene sequence of the labels; $n$ = 594 ants and 77 pellet pools). This allowed us to discriminate between successful pathogen removal (spores collected into the ants' infrabuccal pockets or expulsed as pellets), unremoved infectious spores (remaining on the body of the spore-treated individuals), and transmitted spores indicative of cross-contamination (found on the other ants' bodies; Supplementary Fig. 1) that had taken place over the course of the 90 min after treatment.

## Pathogen exposure triggers individual and collective hygiene

Ants treated with the pathogen intensified selfgrooming activity[24,33] (Fig. 1b, Supplementary Table 2), and the oral uptake of self-produced formic acid, their antimicrobial poison (Supplementary Fig. 2, Supplementary Table 2), which aids spore disinfection in the infrabuccal pockets[37]. Any type of treatment (F, f, C) – not only pathogen application – reduced the propensity of an individual to groom others, suggesting that treatment per se was experienced by the ants as a disturbance (Fig. 1c, Supplementary Table 2). Therefore, treated ants engaged in selfgrooming in relation to the spore load they received, but equally refrained from grooming others as a result of the general disturbance they experienced by being treated. The nestmates, on the other hand, provided high levels of sanitary caregiving, which they directed mostly towards pathogen-treated individuals, and less frequently also towards control-treated individuals. This is consistent with the observation that sanitary care in ants and other social insects like termites is triggered by chemical pathogen cues detected on the exposed individuals, such as the fungal membrane compound ergosterol[38,39].

The untreated nestmates also increased selfgrooming as a response to pathogen exposure of the group (Fig. 1b, Supplementary Table 2). In particular, they spent more time grooming themselves after grooming a spore-loaded (F,f) individual than after grooming another untreated nestmate (N), whereas grooming a control-treated ant elicited a context-dependent response (Supplementary Fig. 3a). Notably, nestmate selfgrooming was triggered only by performing grooming towards – but not by receiving grooming from (Supplementary Fig. 3b) – infectious ants, and was therefore not a simple reaction to contact. In contrast to treated individuals who showed an increase in the use of their sanitizing poison, the nestmates decreased the utilization of their poison, which is costly to produce[40] (Supplementary Fig. 2, Supplementary Table 2). It is still unclear whether nestmates refrain from using own poison when sensing the increased application of the volatile formic-acid-rich disinfectant by the treated individuals, or whether they can assess their low risk of getting cross-contaminated with high, disease-causing pathogen levels[41]. Independently of the underlying mechanism, such increased self-hygiene by nestmates of infectious individuals is predicted by epidemiological modeling to evolve, as it reduces disease risk for the colony[33].

## Informed grooming choices based on spore load dynamics

Together, individual and collective sanitary behaviors led to a >80% spore reduction on the exposed individuals from the initially-applied spore load. 85% of nestmates were spore-contaminated at the end of the experiment, albeit at very low pathogen levels (Fig. 2a, see Methods) that rarely cause disease but can trigger protective immunization[41]. The number and type of spores retrieved from each ant's infrabuccal pocket at the end of the experiment (as quantified by fluorescence-specific ddPCR) correlated well with the ant's grooming activity in the last sixty minutes of the experiment (min 30–90 after exposure). Earlier grooming events (min 1–30), however, were less well reflected in the stored spores, likely because the ants had already expelled these spores as a pellet in the meantime (Fig. 2a, Supplementary Fig. 4). Experimental groups that initially received an overall higher spore load produced higher numbers of pellets, yet the number of spores packed into each pellet was independent of treatment (Supplementary Fig. 5a). This suggests that the ants groom until their infrabuccal pockets have reached a particular filling state, which then triggers pellet expulsion. Importantly, the effective grooming time needed until pellet expulsion increased over the course of the experiment, implying a reduced efficiency of spore removal as the spore load in the group decreased (Supplementary Fig. 5b). This observation was quantitatively captured by a Type II functional response model[42] for grooming efficiency (mathematically equivalent to a Michaelis-Menten reaction kinetics; Supplementary Table 3, Methods), in which the spore removal rate during grooming is at saturation when the pathogen load on the infectious ant is high, but decreases at lower pathogen loads. Fitting this model to our spore data allowed us to infer the current load of each spore-treated ant at any time during the experiment, by back-computation from its remaining spore load and the grooming it had received (Fig. 2b).

For every decision in which a nestmate chose to groom either of the two spore-treated individuals ($n$ = 5001 decisions of 196 N in 49 FF, Ff, ff treatment groups), we used this time-resolved inferred spore load information to determine if the nestmate targeted the individual with the currently higher load (hereafter higher-load individual) or not. Remarkably, we found that *L. neglectus* ants appear capable of making informed grooming choices, by estimating the instantaneous spore load on other ants and biasing their grooming towards the higher-load individual, where grooming is most effective. In detail, the ants targeted the higher-load individuals more frequently (Fig. 2c, see also Supplementary Fig. 6), while not modifying the duration per grooming event (Fig. 2d). Notably, this grooming bias towards higher-load individuals relied on the ants' current, but not their initially-applied, spore loads, as individuals with higher initial spore load were only preferentially targeted early in the experiment (Supplementary Fig. 7), when the initial load still approximates the current load (Fig. 2b). The ants are thus reacting to constantly-updated pathogen load information when making their grooming choices.

## Probabilistic modeling identifies individual decision rules

We next sought to turn these statistical observations into a consistent and sufficient set of behavioral rules capable of mathematically predicting moment-by-moment individual ant decisions and the emergent colony-level dynamics. We developed a class of models[32] in which individual ants stochastically switch between discrete behavioral states of selfgrooming, allogrooming, or not performing any sanitary action (Supplementary Note 1, Fig. 3a). Cross-validated model selection identified the factors that ants integrate to choose whether to groom in the next time instant or not (Supplementary Note 1). Our model recovered the observed grooming activity best when predicting it based on the combined information about the spore load that the ant had encountered on others within the past -minute (L), and the grooming it had received by others within the past -10 seconds (R, Fig. 3b). While the combined model using R and L factors improved global prediction error shown in Fig. 3b by a small amount compared to the L-only model, it generated a significantly larger improvement in moment-to-moment predictability in the choice of next ant behavior

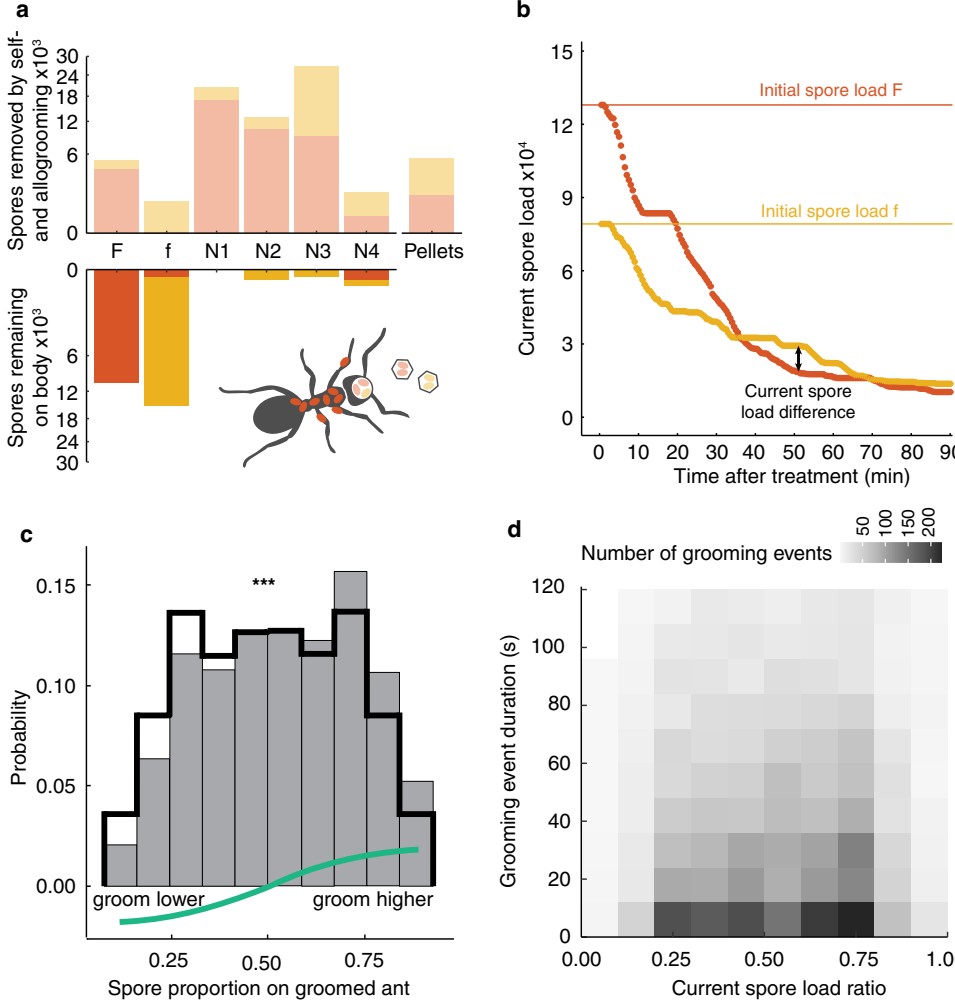

**Fig. 2 | Sanitary care behavior depends on pathogen load. a** Ff-treatment group example of measured final spore loads remaining on each ant's body or acquired by transmission (opaque), and removed (translucent color) by collection into the ant's head or as disposed pellets. Red indicates spores originally applied to the F-individual, yellow to the f-individual. **b** Inferred spore load dynamics for the two pathogen-treated ants in panel **a**. Horizontal lines show back-computed initial loads of the F- and f-individual; arrow depicts exemplified current spore load difference. **c** Distribution of the proportion of the spore load on the groomed individual (out of total spores on both spore-treated ants), assembled across all nestmate allogrooming events (gray bars), when compared to chance (black line) reveals the ants' preference to groom higher-load ants (bootstrapped Kolmogorov-Smirnov test, $D = 0.053$, two-sided $p = 1.632e^{-6}$ (shown by *** as ≤0.001), green line depicts smoothed observed to expected-by-chance difference; see also Supplementary Fig. 6). **d** Duration of individual allogrooming events (grooming events of duration <2 min [90% of events] depicted) does not systematically depend of the current load proportion (42/45 pairwise Kruskal–Wallis tests two-sided adjusted for multiple testing $p > 0.05$); **c**, **d** based on $n = 196$ N from the 49 FF, Ff, ff replicates. Source data are provided as a Source Data file.

(Supplementary Note 1). To replicate the observed ant behavior, our model had to further include an individual disturbance factor (ρ, Fig. 3c) describing the change in ant behavior upon receiving any kind of treatment per se (independent of its infectiousness level).

Finally, our analysis revealed that a preference of an ant to groom the higher-load individual (Fig. 2c) is most parsimoniously explained by what we call a sequential-choice rule: an ant probes one group member after the other, and commits to groom the currently probed ant with a probability that increases with the probed ant's current spore load (Fig. 3d; Supplementary Note 1). Notably, this choice rule does not require an ant to compare (and remember) the spore load on multiple other ants. While the empirically identified rule may not be optimal in terms of the total spore removal efficiency as a consequence of its local and sequential nature, it nevertheless leads to a systematic bias towards preferentially choosing higher-load individuals; similar collective choices that do not require individual comparisons (and thus global knowledge) have been reported in other contexts, like nest choice in ants[14,18,43]. Simulations using the sequential-choice rule matched data better than various alternatives (Fig. 3d). The most naïve alternative we

explored was one in which ants choose their grooming targets uniformly at random; this alternative, however, provided the worst match to data. Next, we explored two less trivial alternatives in which ants choose their grooming targets based on information that was not dynamically updated. In the first no-updating alternative, ants did perform sequential choice, but using initial spore loads only; removing dynamic load-updating thus substantially worsened fit to data. In the second no-updating alternative, ants used fixed (but non-uniform) probabilities to choose amongst their F, f, C, N targets. Even when these probabilities themselves were fitted to maximize agreement with data, this alternative still underperformed sequential choice. The last alternative we considered allowed for dynamic updating and assumed that ants could acquire complete information about all current loads, to deterministically pick the ant with highest current load to groom (maximum rule). While such a hypothetical rule would result in the most rapid removal of spores from the colony, it was also not supported by data.

Predictive performance of various rules for grooming choice (Fig. 3d) clearly identified the importance of continuous information

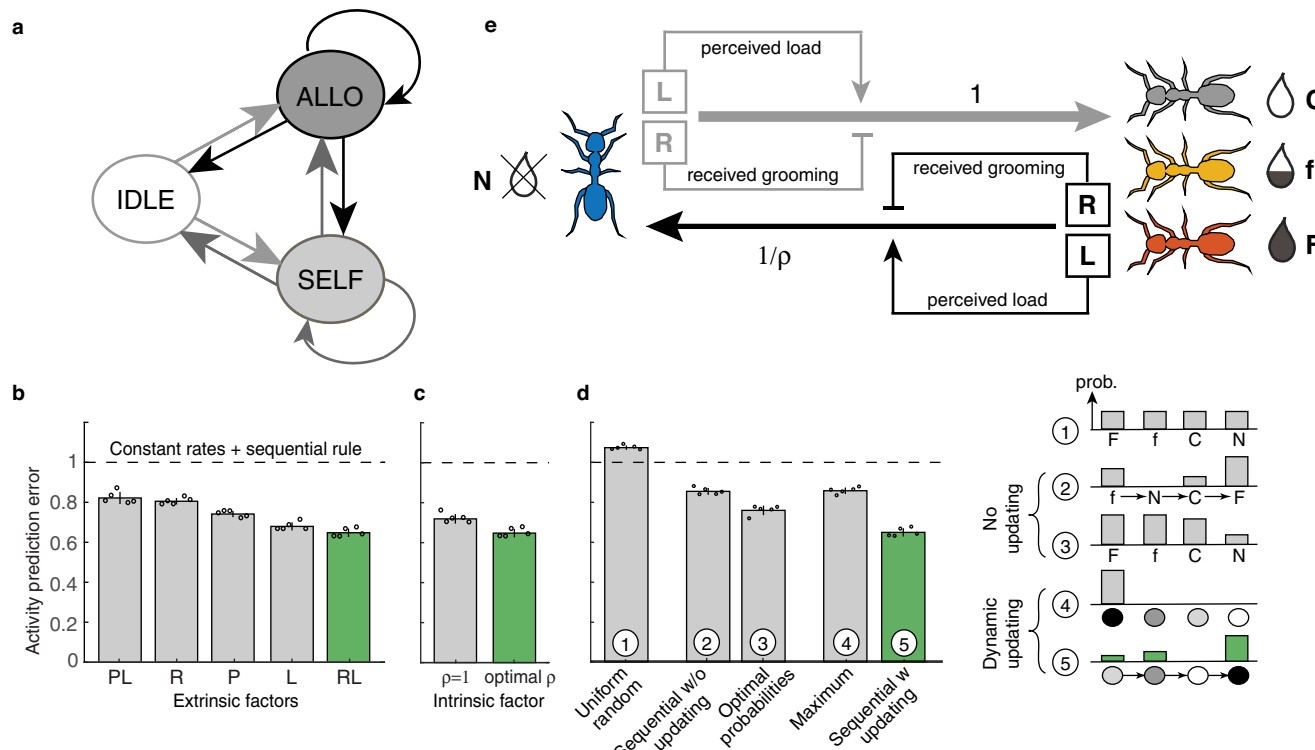

**Fig. 3 | Stochastic model of individual ant decisions identifies key factors that modulate and direct allogrooming activity. a** At each moment, an ant is either inactive (IDLE state), selfgrooming (SELF) or grooming another ant (ALLO). Stochastic transitions between states (arrows) depend on a range of factors to be identified. (**b**) Model selection identified predictive factors that the ant experienced in the recent past, such as performed (P) and received (R) allogrooming, and encountered spore load (L), by minimizing prediction error on 5 independent replicate simulation sets for the time-resolved activity across all ant classes (F,f,C,N) and treatment combinations. The best model (green) uses factors R and L, **c** an individual disturbance factor ($\rho$) by which all treated ants (F,f,C) equally suppress their allogrooming compared to untreated ants, as well as **d** a sequential-choice rule with dynamic updating of spore load information (rule 5, green bar) to pick a target for grooming. This rule is favored by model selection (bars at left) over alternative rules (rules 1-4; all rules schematized at right, see Supplementary Note 1

for details). In alternative rule 1, ants pick grooming targets uniformly at random. Alternative rule 2 is a variant of the sequential-choice rule with the same parameters as its dynamic counterpart (green bar), but with loads being the initial rather than dynamically-updated current loads. Alternative rule 3 uses static probabilities for picking each ant depending on its treatment; probabilities have been optimized for best fit to data. Alternative rule 4 assumes each ant has access to global current load information to deterministically groom the ant with the currently maximal load (circle darkness reflects spore load intensity). In b-d, bars represent the mean of the 5 individual simulations (each depicted by a circle; error bar shows ±std), relative to the constant rates model (dashed horizontal line). **e** Schematic of the identified best model. Ants amplify allogrooming when recently having perceived high spore load on others, and suppress it after having received grooming. Transition to allogrooming is additionally suppressed ($\rho$) in all treated ants.

updating and suggested sequential choice as a biologically-plausible procedure by which such information could be utilized by individual ants. Sequential choice predicts a clear experimental signature: the existence of multiple transient ant-ant interactions that precede, but do not immediately lead to, allogrooming. We therefore analyzed in detail the ants' behavior in two of our experimental replicates (Supplementary Fig. 8) to see if the ants may sequentially probe their targets before committing to groom, preferentially, the higher-load ant. Indeed, we found bouts of antennation – a common recognition and discrimination behavior in ants[44] – preceding most allogrooming events, whereby an ant would make several transient contacts with different target individuals before finally choosing an ant to groom; typically, the chosen ant was also antennated immediately prior to grooming. While our experiments do not permit us to unambiguously and causally identify antennation as the probing mechanism underpinning the sequential choice, they provide a possible mechanistic basis and correlational evidence in support of this idea.

Taken together (Fig. 3e), an ant is more likely to become an allogroomer when (i) it recently perceived pathogen load *on others* – indicating an ongoing colony-level infection; (ii) it was recently not a recipient of grooming *by others* – indicating that it is likely not highly contagious itself; (iii) it did not experience any kind of disturbance (e.g., treatment) during the experiment. Furthermore, when an ant

becomes an allogroomer, (iv) it preferentially targets higher-load ants. We identified how ants combine information (i-iv) by maximum-likelihood inference of model parameters (Supplementary Note 1), using data on ant moment-by-moment decisions. The resulting model simulations accurately captured the sanitary behavior and spore removal across all treatments and replicates, both at the group and individual level, across the entire hour-long experiment (Fig. 4; Supplementary Note 2). Correct prediction of long-term dynamics from momentary individual decisions constitutes a non-trivial test of the model[45] and parallels the analysis of collective dynamics on long timescales in statistical physics of active matter[46].

### Individual decisions allow efficient whole-colony spore removal

The inferred rules that best capture ant behavior are only based on local information that the ants can acquire from their close contacts. Theoretically, the colony could remove even more spores if the ants always groomed the individual with the currently highest load (maximum rule, Fig. 3d; Supplementary Note 1). This consideration, however, ignores biological constraints faced by individual ants in large collectives that they form in their natural colony conditions[47]. It would require each ant to assemble global information by assessing and remembering the maximum pathogen load of its group members, to subsequently cognitively identify and physically locate the most infectious individual to

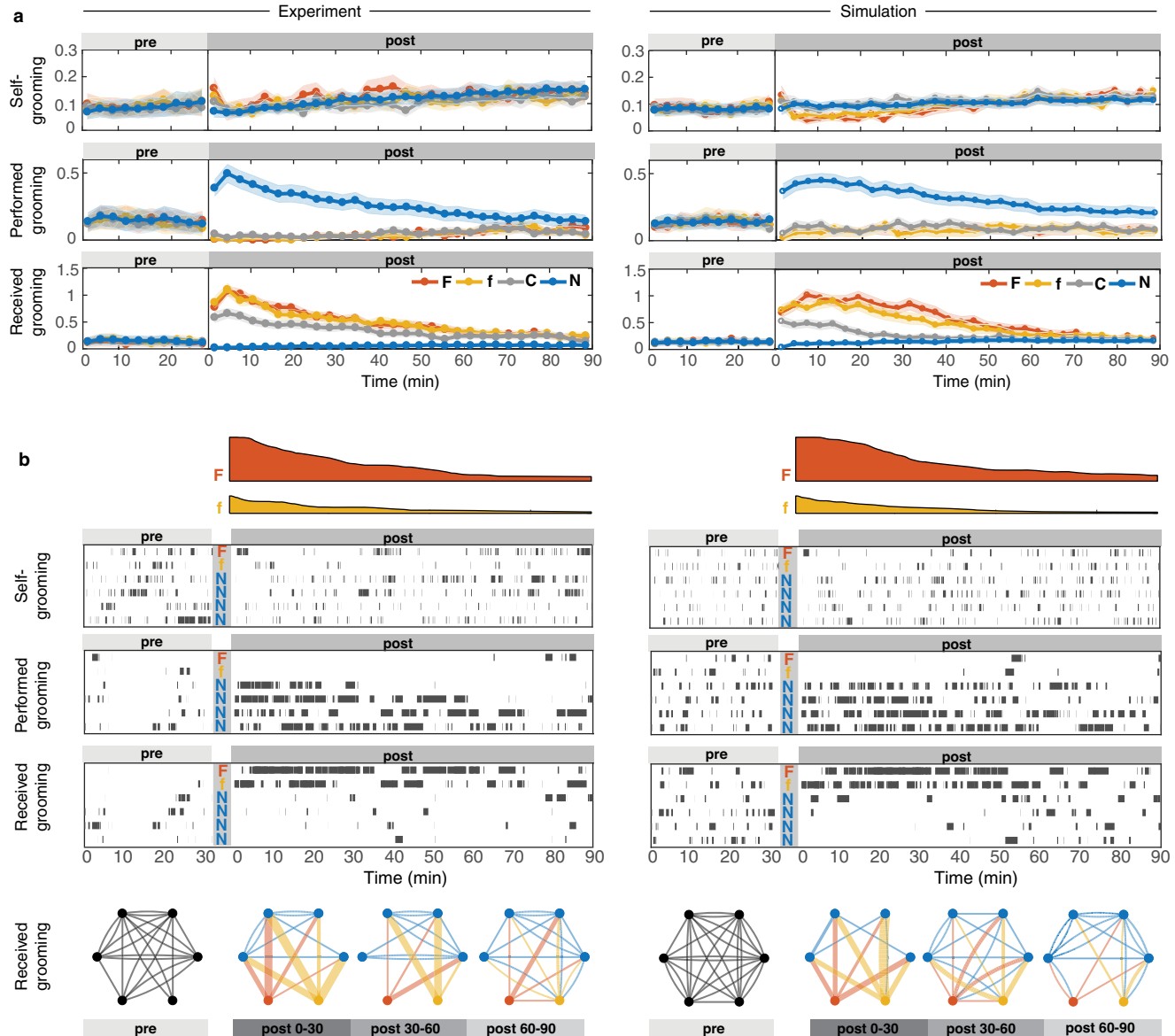

**Fig. 4 | Stochastic model of individual ant decisions predicts sanitary care behaviors across the length of the entire experiment.** Side-by-side view of experimental data (left) and stochastic simulations (right), aggregated and exemplified for individual ants. **a** Averaged activity traces binned into 30s windows. Circles depict the means, shaded areas the sem centered around the means, which are magnified for better visibility in the plot by a magnification factor of 1.5x for the treated [F, $n = 66$, red; f, $n = 65$, yellow; C, $n = 67$, gray] and 3x for the nestmate [N, $n = 396$, blue] ants of the 99 replicates), partitioned by grooming type

(selfgrooming, performed and received allogrooming) and by ant treatment, shown pre- and post-treatment. **b** Detailed activity rasters for each ant (F, f, and 4N as rows within each raster for a Ff example). (Top) Inferred current spore load on F- and f- ants decreases due to grooming in data and simulation on a comparable timescale. (Bottom) Grooming networks for the four 30-min intervals of the experiment (pre: 30 min pre-treatment period; post-treatment period separated into three 30-min periods: post 0–30, 30–60, and 60–90 min after treatment; edge thickness = total duration of received allogrooming events per ant).

groom. We explored the efficiency of different choice rules in a separate set of stochastic simulations where individual ants committed their time to either first probing other ants to gather information on their pathogen load state or to directly allogroom encountered others, thereby reducing their pathogen load. As soon as probing other ants incurs any time cost, our simulations (Supplementary Note 2), show that the experimentally-motivated sequential-choice rule (based on cheap partial information) will outperform the hypothetical maximum rule (based on costly complete information) as colony size increases (Fig. 5), in a classic manifestation of the exploration-exploitation tradeoff[48,49]. Here, exploration refers to the effort by nestmates to locate the ant with highest spore load, where grooming (exploitation) would be most efficient. This is in analogy to the application of the same tradeoff to the problem faced by foraging animals that need to balance the time spent

on searching for new and possibly rich foraging grounds, with the time spent exploiting known, but perhaps more meagre, grounds. We point out that the decisions faced by the nestmates could also be understood in terms of the speed-accuracy trade-off[50–52]. Here, nestmates need to invest more time (i.e., to probe colony members) to make a more accurate choice (i.e., to locate the highest-load individual and deliver most efficient care). While the exact dependence of grooming probability on the observed load in the sequential-choice rule that leads to most efficient spore removal depends on various factors, all efficient rules share a very small probability of targeting a non-infectious ant for grooming (Supplementary Note 1, Supplementary Note 2), as we observe in the data. We note that even in small groups as used for our experiment, where ants could conceivably gather global information and use the optimal maximum rule without much efficiency cost (Fig. 5), our data

preferentially support the sequential-choice rule. This suggests that the ants' decision about whom to groom reflects a hard-wired optimal strategy that evolved under the selection pressures in natural colony sizes.

### Grooming decisions reflect load differences and social feedback

Next, we explored the biological implications of our model insights. We found that the grooming bias of nestmates towards the higher-load individual is more pronounced when the two treated individuals clearly differ in load, but is already expressed when the difference

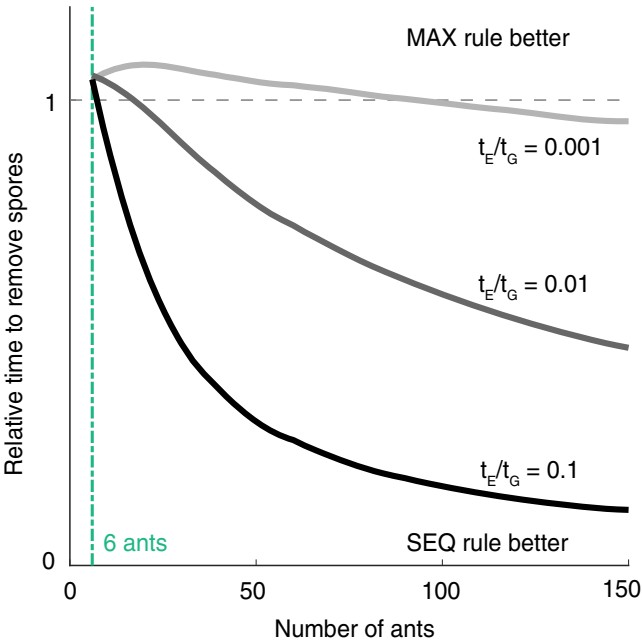

**Fig. 5 | Illustration of the exploration-exploitation tradeoff in a simplified model of ant decision making.** Ants incur a time-cost $t_E$ per encounter to estimate the spore load on a target ant (exploration) and $t_G$ to groom it (exploitation). The inferred sequential-choice rule (SEQ, partial information) outperforms the maximal rule (MAX, complete information) as the colony size grows for any nonzero $t_E/t_G$, in terms of time needed to remove 90% of pathogen.

between the two individuals is small (Fig. 6a; $n = 8129$ grooming decisions by 328 N in the 82 replicates with at least one spore-treated individual). This allows nestmates to preferentially groom the higher-load individual, even if the two individuals initially received equal spore loads, which subsequently diverged due to differential self- and allogrooming history (Supplementary Fig. 9). Such a behavior will inevitably emerge when ants use information on spore loads they perceived in the very recent past (Supplementary Note 1), rather than keeping long-term memory of the initially-applied load (Supplementary Fig. 7). Such constant updating of pathogen load information appears key for the ants to react dynamically to changes in disease risk.

Our model also uncovered that ants respond to social feedback when making their sanitary care decisions (Fig. 3b). Social encounters have been known to affect various behaviors, e.g. foraging[53,54] and nest evacuation[55] decisions. Here, we found that an ant suppresses own performance of allogrooming after being groomed by others. This suppressive effect is strongest in the first minute after received grooming; its time-course follows an exponential decay with a ~half-minute timescale. Since an ant preferentially receives grooming when its pathogen load is higher, higher-load individuals will be inhibited more strongly by this social feedback. Grooming suppression is a general response to received grooming – it occurs in all ants even in the absence of a pathogenic threat (i.e. in the pre-treatment period). The suppressive effect increased ~2-fold compared to the pathogen-free pre-treatment situation for the treated ants, while it was reduced ~2-fold for the untreated nestmates (Fig. 6b). Untreated nestmates were thus less reactive to the suppressive effect of being groomed after pathogen exposure of their group members. For all treated ants, the strength of the suppression was equally high after any treatment, be it high or low pathogen exposure or control treatment (Supplementary Fig. 10). Thus, any form of disturbance that an ant experiences makes it more reluctant per se to groom others ($\rho$), as well as more responsive to the social feedback it receives from others (R).

The social feedback that we identified in the regulation of collective sanitary care likely allows the ants to continuously evaluate their own pathogen load and hence the infectious risk they pose to the colony. A necessary condition for this mechanism to function is the ability of ants to preferentially groom higher-load individuals: without this ability (e.g., using a uniform random choice rule; Fig. 3d) such a social signal would carry zero useful information and would have no clear adaptive value. Preferential grooming of highly-infectious individuals and social feedback thus provide a compelling example of two

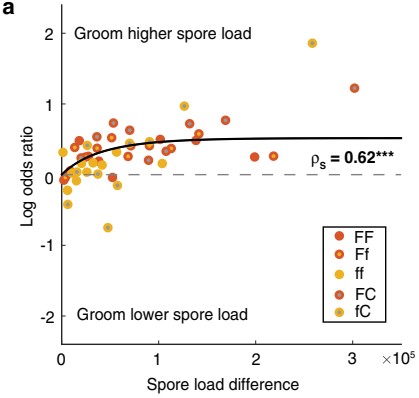

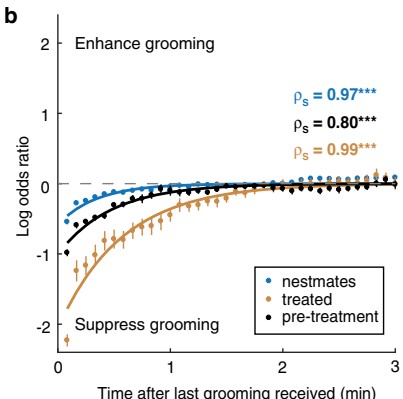

**Fig. 6 | Grooming decisions depend on spore load difference and social feedback. a** Evidence for grooming bias towards the higher-load ant. Nestmates' preference to groom the higher-load ant increases with the difference in current spore load between two treated ants towards saturation (data bins based on 328 N from 82 replicates (except CC); Spearman-rank correlation; black line: theoretical expectation; see also Supplementary Fig. 9). Two-sided $p$-value $p = 1.515e^{-6}$ (depicted by *** as ≤0.001). **b** Evidence for social feedback. An ant's transition to allogrooming is suppressed if it received grooming in the recent past (pre-treatment: all ants black, $n = 594$; post-treatment: nestmates blue, $n = 328$; treated ants brown, $n = 164$; for separate F,f,C see Supplementary Fig. 10). Time-dependence assessed by Spearman-Rank correlation (rho given; mean ± std over 5 s bins; solid lines display exponential fits), two-sided $p$-values: nestmates $p < 1 e^{-11}$, treated individuals $p < 1 e^{-11}$, pre-treatment: $p = 2.1 e^{-7}$ (all depicted by *** for ≤0.001). Source data are provided as a Source Data file.

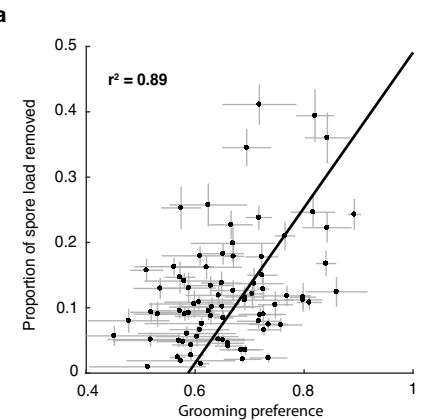

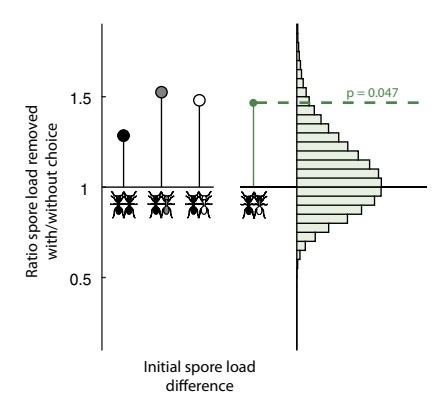

**Fig. 7 | Grooming choices biased towards higher-load individuals result in efficient pathogen removal. a** Groups in the main experiment in which nestmates groomed the higher-load individual more (high model-predicted grooming preference, mean ± std over 5 simulations per replicate) remove more spores by nestmate allogrooming and pellet formation (measured means ± std using estimated detection error; $n = 82$ replicates; $r^2$ of linear functional model given; see also Supplementary Fig. 11). **b** Functional knock-out experiment in which ants were prevented from choosing (i.e., nestmates only faced one treated ant) compared to a choice situation (i.e., nestmates had two treated ants to choose from). The ratio of the total removed spores (collected in nestmate heads and expelled as pellets per group, $n = 98$ groups, 14 each of 4 free-choice and 3 no-choice situations; green dot at right shows the mean across different choice scenarios at left) is higher in free-choice compared to the matched no-choice situations (comparison to the null distribution in pale green, calculated by bootstrapping, $n = 10^5$, one-sided $p = 0.047$; see Supplementary Notes 3). Importantly, all choice situations – difficult (two spore-treated individuals, both with high initial load; two black ants), moderate (two spore-treated individuals with different initial spore load; black plus gray ant), or easy choice (only one of the two individuals treated with a high load of spores, the other being control-treated; black plus white ant) – show similar ratios of spore removal with vs. without choice. Source data are provided as a Source Data file.

behaviors that combine synergistically. An appealing hypothesis is that social feedback, especially when reinforced by continued grooming or by multiple ants grooming the same individual simultaneously, could be more accurate than any direct self-assessment of the individual's own pathogen load. This could constitute a reliable mechanism for the ants to adjust their sanitary caregiving to own infectiousness, thereby reducing the epidemiological risk of spreading the pathogen[33].

**Informed grooming increases group spore removal efficiency**
We quantified how grooming decisions affected a group's spore removal efficiency, that is, the proportion of spores that the ants removed in the 90 min after exposure from the treated ants by collecting them into the nestmates' infrabuccal pockets and inactivating them as pellets. We found that groups which had more frequently successfully targeted the higher-load individual (better choosers) removed more spores than groups which less frequently picked the higher-load ant (worse choosers; Fig. 7a, Supplementary Fig. 11). We next evaluated how a complete prevention of choice would affect spore removal efficiency, by performing a separate functional knock-out experiment. We compared spore removal in groups that had free choice (groups of 4 nestmates with two treated individuals, as in the initial setup) with spore removal in other groups that did not have free choice (initial groups split up into two halves, each with two nestmates and one treated individual only; $n = 98$ groups of total 462 ants, 14 replicates each of 4 choice and 3 no-choice situations, see Methods). We found that the total spore removal in the no-choice scenario, i.e., spore removal summed over the two split half-groups, never reached the spore removal levels of the corresponding unsplit freely-choosing group, independently of the initial spore load difference between the two treated ants (Fig. 7b).

Taken together, the observed higher spore removal of better choosers (Fig. 7a), as well as the observed higher spore removal of groups with choice (Fig. 7b), indicate that many small grooming biases in individual ant decisions accumulate to a functional benefit at the collective level. The ability of individual ants to preferentially target higher-load individuals (rather than choosing uniformly at random; Fig. 3d, Supplementary Note 1), combined with the higher grooming

efficiency at higher spore loads (Type II functional response model, see above), builds up to highly efficient pathogen removal at the group level. This effect is already detectable for our experimental groups of only six ants, in line with previous reports that collective benefits can emerge already at these very small group sizes[36].

## Discussion
In this study, we extracted individual decision-making rules underlying collective hygiene in ants. These rules are cognitively simple, scalable, and plausible, since they only require short-term memory, contact-based social feedback, and locally-accessible pathogen threat information. Despite these limitations that exert their effect at the level of individual ants, the identified rules interact synergistically and amplify into efficient pathogen removal with expected epidemiological benefits at the colony level. Below we rationalize how this synergy comes about.

The first essential ingredient is the dynamic and continuous re-assessment of current pathogen loads. Even if constrained by limited individual cognitive abilities, this dynamic updating is key for the ants to react quickly and appropriately: the pathogen load and hence transmission risk are not fixed, but change continuously as a consequence of the ants' sanitary actions themselves, requiring continual information updating. Second, the fact that ants target individuals with higher current pathogen load for grooming with higher probability, coupled with the mechanistic aspects of grooming that permit higher spore removal rates at high pathogen loads, combine into a very efficient pathogen removal for the entire colony. Third, whilst pathogen load perceived on others is an excitatory factor promoting grooming, own pathogen load inhibits grooming performance, thereby limiting the spread of infection[33]. Own pathogen load seems to be assessed by the ants indirectly, requiring social feedback by others in the form of received grooming – which must be targeted preferentially towards the infectious ants for this mechanism to work. Finally, we found that the ants become more reactive to socially-acquired information when they also individually acquire information on their own disturbance, reiterating the relevance of combined internal and external information integration in social insects[55,56].

Taken together, efficient collective hygiene in the colony emerges from mechanistic amplification of behavioral choices, interplay of excitatory and inhibitory factors through feedback, as well as individual and social information processing. The identified rules contribute to preferential grooming of the most-infectious colony members by the individuals with the lowest own infectiousness, identified by epidemiological modeling as important to reduce disease risk for the colony[33]. Such immediate functional benefits – and thus evolutionary relevance – of emergent behaviors can rarely be pinpointed[57], despite recent technical advances that spurred a multitude of studies on collective actions across the social, physical, and life sciences[58,59]. Our results showcase the importance of collective behaviors for the evolution of social insects, whose fitness – similar to other cooperative entities like cells in a body – inherently depends on group-level performance[60-62].

## Methods

### Ant host and fungal pathogen
As host species, we studied the invasive garden ant, *Lasius neglectus*. We collected several hundred workers, multiple queens and brood of this species from its introduced supercolonial population in the Botanical Garden in Jena, Germany (N 50° 55.910 E 11°35.140)[63] in June 2015 and September 2022 and reared them in the laboratory with sugar water and minced cockroaches. Experiments were performed with workers sampled from worker chambers inside the nest, to avoid both old foragers and freshly emerged callows, in a humidity- and temperature-controlled room at 65% RH and 23 °C. Collection of this unprotected species and all experimental work followed European and Austrian law and institutional ethical guidelines.

As pathogen, we used the obligate-killing entomopathogenic fungus *Metarhizium*, a common natural pathogen of ants[64] including *Lasius*[65], in particular the infectious conidiospores (here also abbreviated as spores) of the *Metarhizium robertsii* strain ARSEF 2575[66], with either an integrated green (eGFP) or red (mRFP1) fluorescent label (obtained from M. Bidochka, Brock University; labels abbreviated as GFP and RFP). GFP- and RFP-labeled spores did not differ in their virulence to the ants.

### Sanitary care experiment design and setup
We observed all ant behavior and quantified pathogen removal and transfer in 99 experimental groups, each consisting of six individually color-coded workers (with an Edding 780 dot applied to each worker 18–24 h before experimental start) in plastered petri-dishes (Ø 35 mm, SPL Life Sciences) with glass covers (Ø 51.5 mm, Edmund Optics). Four workers remained untreated, and are referred to as nestmates (N). The other two were chosen randomly after an initial 30-minute filming period (pre-treatment period) to be treated with one of three treatments: an F-treatment of a high dose of the fungal pathogen, an f-treatment of a low, i.e. half of the high, fungal dose, or a C-treatment consisting of a pathogen-free control treatment. The all-pairwise combinations of these three individual ant treatments resulted in six different group treatments (FF, Ff, ff, FC, fC, CC; Fig. 1a). To allow for distinction of the spore origin in groups with two pathogen-exposed ants (FF, Ff, ff), one worker received the GFP-labeled spores, the other the RFP-labeled ones, in a randomized manner (Supplementary Table 1). Treated ants were put back into their group immediately after treatment and the behavior of all ants was filmed for another 90 minutes (post-treatment period), resulting in a total of two hours of filming. Filming was performed in a four-camera parallel setup, where the treatment group of the four replicates filmed at a time was randomly assigned, resulting in a balanced design with treatment groups distributed over the course of the day, over a total of six recording days (rolling-shutter cameras from IDS UI-1640LE USB 2.0 CMOS, 15 fps, 1024 × 1024, 1.3MPixel, 1.3" Aptina Sensor, Rolling-shutter; fixed focal length lens 6MM 1/1.8" f 1.4-f/16 C-mount, Edmund Optics;

Streampix 5 digital video recording software [NorPix, Inc.] for acquisition). After filming, all ants and the spore pellets they had disgorged, were frozen at −80 °C for later pathogen load quantification. From the original 108 replicates (18 per treatment group), 9 had to be excluded for technical errors occurring during experimental performance, so that we obtained 16 to 17 replicates for each of the six treatment groups (*n* = 99 replicate groups of six ants, total 594 ants, of which 66F, 65f, 67C, and 396N; Supplementary Table 1).

### Pathogen exposure
The infectious conidiospores of *M. robertsii* were harvested in sterile Triton X-100 (Sigma; 0.05%, diluted in milliQ water) from 6.5% sabouraud dextrose agar plates freshly before the experiment and their germination was confirmed to be >95%. Pathogen-treated workers were individually exposed to a 0.3 µl droplet of the fungal spore suspension at a concentration of $1 \times 10^9$ spores per mL for the F-workers and half of this concentration, $5 \times 10^8$ spores per mL, for the f-workers. C-workers received a control treatment of the same volume of sterile Triton X-100 only. To this end, each ant was held in forceps and its abdomen was gently drawn through the droplet of the respective treatment suspension, placed on a clean glass slide, until the droplet was completely taken up. Spore quantification (as detailed below) directly after exposure revealed that this treatment led to an effective application of $2.00 \times 10^5$ spores (median; interquartile range IR $1.36–2.55 \times 10^5$; *n* = 30) for the F-workers and of $1.04 \times 10^5$ spores (IR $0.88–1.31 \times 10^5$; *n* = 30) for the f-workers.

### Behavioral quantification
We analyzed the individual and social behavior of each of the 594 ants for the total 2-hour duration of the experiment using a behavioral-logging software (Solomon Coder v. 17.0[67]), which allows scoring with a frame-based resolution of the start and end frame of each event. As ant treatment had been applied to two individuals that were randomly chosen independent of their color, no association could be made by the observer between ant color and treatment, neither within nor across replicates and treatment groups. Ant color-to-treatment assignment was also not revealed in the videos, ensuring bias-free behavioral scoring. We determined event number and duration of (i) selfgrooming behavior of the ant's own head or body, (ii) allogrooming performed towards group members, (iii) allogrooming received by group members, (iv) poison-uptake behavior from the acidopore into the ant's mouth, used for antimicrobial treatment by the ants[37], and (v) pellet disgorgement of spore pellets. These contain the spores collected in the infrabuccal pocket during grooming, compacted and expelled as pellet. We also scored other behaviors, which we did not include in the analysis due to their short duration (1 frame, equivalent to 1/15 s for antennal strokes) or their rare occurrence (food exchange behavior, trophallaxis). We did not observe the ants performing any aggressive behaviors. We further observed the first 30 min after treatment of two ant groups (one Ff and one FC example) in detail for their antennation behaviors (recorded as events due to their short duration), to set these non-grooming encounters into context with the grooming encounters.

### Pathogen quantification
After the end of the experiment, i.e. 90 min after treatment of the treated individuals, we quantified the number of spores of each of the 594 ants, as well as the spores packed into the pellets produced by the ants. The pellets produced by each group of ants were pooled for spore quantification, resulting in *n* = 77 pellet pools (no pellets were produced in any of the 17 CC replicates, in three fC replicates and one FC replicate, and one sample was lost before quantification). Moreover, we quantified the spore number of an additional 75 ants, which were not used in the experiment but frozen immediately after application of the F- or f-dose (*n* = 30 each, of which 15 using GFP- and

15 using RFP-labeled spores) and a control treatment with Triton X-100 (C; $n = 15$), in order to determine the initially-applied spore load of the treated ants at the start of the experiment. GFP- and RFP-usage was randomized across ant treatments, colors and replicates and sample labels did not contain treatment information.

After freezing, we removed the head of each ant with a clean scalpel and separately processed its head and body, to obtain distinct estimates of (i) the spores collected during grooming in the infrabuccal pocket inside each ant's head, where they are chemically disinfected using poison the ants took up before being expelled as pellets[37], and (ii) the spores remaining on the ant's body surface, representing unremoved spores from the body of the spore-treated ants, as well as contamination of the nestmate body by cross-contamination (Supplementary Fig. 1; see also below). For each of the 1415 samples (1338 ant dissection samples and 77 pellet pools) we simultaneously obtained absolute counts of the GFP- and RFP-labeled spores, by a multiplex droplet digital PCR (ddPCR) assay.

To this end, all samples were homogenized in a TissueLyser II (Qiagen) using a mixture of one 2.8 mm ceramic (VWR), five 1 mm zirconia (BioSpec Products) and approx. 100 mg glass beads (425–600 μm; Sigma), in two steps (2 × 2 min at 30 Hz), as in ref. 21. Total DNA was extracted using Qiagen DNeasy96 Blood and Tissue Kit according to the manufacturer's instructions, with a final elution volume of 50 μl Buffer AE. To perform absolute quantification of the two labeled spore variants simultaneously, we designed a multiplex ddPCR probe assay targeting the single copy mRFP1 (KX176868.1) and single copy eGFP gene sequences (NC 025025.1). Primers and probes were designed using Primer3Plus[68] and have been shown to exclusively amplify the respective gene of interest. The genomic DNA was digested using EcoRI-HF and HindIII-HF enzymes (both New England Biolabs) within the 20 μl ddPCR reaction, comprising: 10 μl of 2x ddPCR Supermix for probes (Bio-Rad), 14 pmol of both eGFP primers (forward: 5′-AAGAACGGCATCAAGGTGAA-3′, reverse: 5′-GTGCTCAGGT AGTGGTTGTC-3′; Sigma), 18 pmol of both mRFP1 primers[69] (forward: 5′-CTGTCCCCTCAGTTCCAGTA-3′, reverse: 5′-CCGTCCTCGAAGTTCA TCAC-3′; Sigma), 5 pmol of eGFP probe (5′-[HEX]CAGCTCGCCGACC ACTACCAGCAGAAC-3′ [BHQ1], Sigma), 5 pmol of mRFP1 probe (5′-[6FAM]AGCACCCCGCCGACATCCCCG-3′ [BHQ1], Sigma), 10 U each of EcoRI-HF and HindIII-HF (both New England Biolabs), 2.8 μl nuclease-free water (Sigma) and 2 μl DNA template. Droplet generation was done using the QX200 droplet generator (Bio-Rad) according to manufacturer's recommendations.

Droplets were transferred into a 96-well plate (Eppendorf) for PCR amplification in a T100 Thermal Cycler (Bio-Rad). Cycling conditions were as follows: enzyme activation for 10 min at 95 °C, followed by 40 cycles of 30 sec at 94 °C and 1 min at 56 °C, followed by enzyme deactivation for 10 min at 98 °C. For the entire protocol, the ramp rate was set to 2 °C / sec. Following PCR amplification, the PCR plate was put into a QX200 droplet reader (Bio-Rad) for the readout of positive and negative droplets. Data analysis was done using the QuantaSoft™ Analysis Pro Software (Bio-Rad, version 1.0). The thresholds were set manually to 3000 for FAM (reporter for mRFP1) and 2000 for HEX (reporter for eGFP). Samples with a total droplet count of <10,000 were repeated. Background noise in the quantification of spores was defined as the maximum number of copies read in the non-target channel (i.e reads in FAM channel for eGFP exposure, and reads in HEX channel for mRFP1 exposure). Values below background noise level (8 copies for mRFP1 and 12 for eGFP) were not considered. Results are given as copies/20 μl well by the software, from which we then calculated the absolute number of spores per sample (e.g. ant head, body, or pellet).

We found that only <20% of the initially-applied spores remained on the body of the spore-treated individuals after the 90 min of the social interaction (F: median 27497 spores, CI 24254 – 32421, $n = 66$; f: median 13710 spores, CI 9141–17120, $n = 65$). In addition, approx. every

second spore-treated ant also contracted low spore numbers from the other treated individual (spore numbers in case of occurred transmission; F: median 84 spores, CI 47–119, $n = 41$; f: 71.5 spores, CI 11.5–98, $n = 28$), equivalent to a fraction of 0.3% (F) to 0.5% (f) of the final load of the spores that still remained from its own exposure. Nestmates contracted the pathogen at a higher proportion. After the 90 min of interaction to the spore-treated, we detected spores on the bodies of 84% of nestmates (275 out of 328 untreated nestmates from the 82 groups with at least one pathogen-treated ant). Again, the number of spores transmitted was very low, with a median of 96 spores (CI 83–102, $n = 275$) per nestmate that had contracted spores. Nearly all nestmates (311/328, 95%) had collected spores in their head, with a median of 1453 spores (CI 942–1836), reflecting their recent grooming activity (Supplementary Fig. 4).

We used a conservative approach to evaluate the error that we introduce by accounting all spores quantified by ddPCR from the head samples of the nestmates as spores collected into the infrabuccal pocket, while some may have been attaching externally to the head capsule. To this end, we set up another two replicates of the treatment group with the highest overall spore load (FF) and analyzed the spore number detectable on the outside of the head capsule of the four nestmates after the 90 min of interaction with the two F-individuals by fluorescent stereomicroscopy (Leica MZ16 FA with Filter Cube: ET DsRed; Software: Leica Application Suite Advanced Fluorescence 2.3.0; as in[41]). Contrasting to the main experiment, we here treated both F-individuals with the RFP-labeled spores, as (i) this label clearly contrasts to the autofluorescence of the ant cuticle, and (ii) we were only interested in the overall number of spores that would attach to the head capsule of the nestmates. We carefully examined the head capsule of the eight nestmates for the presence of fluorescent spores by screening every nestmate head for 30 min. Although we cannot exclude that some spores may have been overlooked, this clearly revealed that the contamination of the nestmate head capsule was restricted to very few spores, e.g. found around the eyes or on the antenna. We did not detect any spores on 3 of 8 nestmates, and those nestmates that had spores had a median of 3 and a maximum of 4 spores. Therefore, whenever we quantified more than 4 spores in a nestmate head sample in our experiment using the ddPCR method, we can confidently assume the spores are inside the infrabuccal pocket. Given that the nestmates across treatment groups had a median of ~1500 spores in their head sample, we consider the noise introduced by our method as very minor.

## Time-resolved current spore load calculation

For each of the spore-treated ants (F, f) we estimated the current spore load contamination on its body with a resolution of 30-s time windows in the 90-min post-treatment period (180 time windows in total) from: (i) the sum of the GFP- and RFP spore counts that remained on its body at the end of the experiment, (ii) the time it selfgroomed its body and was groomed by others, and (iii) the initial distribution of spore loads after F- and f-treatment, as quantified for the 30 F- and f-workers frozen directly after exposure. Given a mathematical model for how the number of spores on an ant could decrease with grooming (spore decay model), one can back-compute the spore load on each ant from its remaining measured spore count (i), and the sequence of grooming events (ii), to any time t during the experiment. If spore loads are back-computed to the beginning of the experiment for all F- and all f-ants, they should recover the experimentally measured distributions (iii). The last fact allowed us to select a best spore decay model and fit its parameters, such that spore loads could subsequently be imputed at any time during the experiment.

To find the best model, we first considered simple models for spore decay, where spores decrease with zeroth-order (constant velocity) or first-order (exponential) kinetics upon grooming. These two simple models either had problematic limiting behavior, or could

not reproduce the measured initial spore distributions. These deficiencies are naturally handled by the spore decay model inspired by Type II FRM, also known as the $h = 1$ Hill function, or Michaelis-Menten reaction kinetics. Here, the ants remove the spores ($S$) over time as $dS/dt = -vS/(S + K)$, with constants $v$ and $K$ to be fitted, where $t$ is the time while the ant is being groomed, where grooming includes both allogrooming (properly accounting for events where multiple ants groom a single target ant simultaneously) as well as body selfgrooming. At high spore loads ($S \gg K$) on the body of the contaminated individual, the removal by the ants by grooming occurs at a maximum rate $v$ (presumably given by the physical limit to how quickly an ant can remove spores). In this regime, the spore number on contaminated ants decreases linearly in time groomed. As spore number decreases further ($S \ll K$), less spores can be captured per unit time, so that the speed of spore removal becomes proportional to the (limiting) current load $S$, and thus the spore number decreases exponentially in time groomed. $K$, known as the dissociation constant in Michaelis-Menten kinetics, gives the transition between the exponential and the linear regime. We fitted the parameters $v$ and $K$ from our experimental data, so that the back-computed initial distributions of spore loads predicted by the model showed the best fit to the initial spore load distributions determined from the quantification of spores of the F- and f-ants directly after exposure (see Supplementary Table 3 for details). The two parameters are chosen to best fit two moments (median and SD) for each of the two distributions (initial load distribution for F and f ants), i.e., two parameters are chosen to best satisfy four constraints. We checked that the fitted values are uniquely constrained by scanning the entire 2D grid in the parameter space and observing a single clear optimum of the fitting loss function. The robustness of $v$, $K$ parameter estimates with respect to the choice of optimization criteria (here, matching the median and standard deviation of distributions) was further corroborated by recovering similar values by minimizing the earth movers' distance between the distributions.

## Experimental prevention of choice

In a follow-up experiment, we tested if preventing the ants from choosing between treated ants would lower their spore removal efficiency, as suggested by the lower spore removal in worse-choosing groups compared to better-choosing groups of ants in the main experiment (Fig. 7a). In this functional knock-out experiment, we either set up ant groups as above, containing 4 nestmates and 2 treated individuals (free-choice situation), or we split the setup in half to consist of only 2 nestmates and 1 treated individual (no-choice situation). In the choice situations, one of the ants was always treated with the high dose (F; $1 \times 10^9$ spores per mL), and the second one either with (i) the control suspension C (easy choice), (ii) a lower spore dose (¼ F or ½ F, i.e. $2.5 \times 10^8$ resp. $5 \times 10^8$ spores per mL, representing a moderate choice difficulty), or (iii) with the same F dose (difficult choice). The no-choice situations contained one ant, treated with either of the three spore dosages (F, ½ F, ¼ F). We analyzed a total of 98 ant groups (14 replicates each of the 7 situations, total $n = 462$ ants, of which 308N). 16–24 h before the experiment, the ants were color-coded as above, yet with all N receiving equal color. To cover the peak grooming activity (Fig. 4a) we froze the dishes 20 min after exposure, and pooled all nestmate heads and pellets per dish for following pathogen load quantification to quantify all spores removed by the group. To this end, we extracted the DNA of the 98 samples as described above, except that annealing temperature was set to 60 °C, thresholds were set to 2200 for FAM and 1800 for HEX and the noise level was 3 copies for both for mRFP1 and eGFP. For the DNA extraction, we ensured equal conditions between the choice and no-choice situations by adding two worker heads from the stock colony to the only two nestmate heads in the 42 no-choice groups to contain same amount of host tissue as the four nestmate heads in the choice situations ($n = 56$).

## Statistical data analysis

Statistical data analyses were performed in R v.3.6.3 and Matlab v.2016b. For every model, we checked model assumptions (i.e. residual normality and heterogeneity, no multi-collinearity, no overdispersion) and influential cases. We built generalized linear mixed models using glmmTMB[70] and lme4[71], and DHARMa[72] as a diagnostic tool. In addition, we used tidyverse[73], forcats[74], data.table[75] and stringr[76] for data formatting, and stats[77] and multcomp[78] for statistical summaries and inference. Effect sizes[79] were calculated using R packages effectsize[80], rcompanion[81] and rstatix[82]. When multiple inferences were made, all significance values were corrected using the Benjamini–Hochberg procedure to protect against a false discovery rate of 5%[83]. Adjusted $p$-values are reported and are all two-sided, with the exception of the choice-prevention experiment (Fig. 7b), which was built on an a priori hypothesis derived from the main experiment. Exact $p$-values given when larger than 1e$^{-11}$. Graphs were made using the R packages ggplot2[84], cowplot[85], ggpubr[86] and scales[87], and Matlab v.2016b (Math-Works).

For each of 594 ants in 99 replicates (Supplementary Table 1), we tested whether its effective time of selfgrooming its body and head, poison uptake, allogrooming, as well as received allogrooming (Fig. 1b–d, Supplementary Fig. 2, and Supplementary Table 2), differed between the pre- and post-treatment period by paired Wilcoxon tests according to individual treatment, i.e. high-load (F), low-load (f), control-treated (C) individuals, and untreated nestmates (N). For the nestmates, we further analyzed if the time they spent selfgrooming was dependent on the treatment of the ant they had groomed, or by whom they had been groomed last (with a max. time window of 3 min before the selfgrooming), by use of Kruskal-Wallis tests followed by posthoc comparisons (Supplementary Fig. 3).

For every grooming event in which an untreated nestmate chose to groom one of the two spore-treated ants ($n = 5001$ individual grooming choices performed by the 196N from the 49 replicates with two spore-treated individuals), we computed the current spore loads on both treated ants, using the Type II functional response model detailed above. For each grooming choice we derived the current spore proportion on the groomed ant as the current spore load of the targeted ant over the total current spore load of both treated ants and compared the observed distribution to a null expectation of uniform random choice (Fig. 2c) by a Kolmogorov–Smirnov test. We further tested for a possible relationship of the duration of grooming events with the spore load proportion on the groomed individual, by binning all grooming events into ten categories of current spore proportion on the groomed individual and counting how many events of particular duration fell into each category, followed by Kolmogorov–Smirnov tests (Fig. 2d).

We tested by logistic regression whether the duration of grooming performed towards the ant with the respectively labeled spores (GFP vs. RFP) predicted the presence of these spores in the head of nestmates from the 82 groups with at least one spore-treated individual (FF, Ff, ff, FC, fC; $n = 328$N, 196 of which had contact to two spore-treated individuals, leading to a total of $n = 524$ data points (Supplementary Fig. 4a). We further performed Spearman-rank correlations between the number of spores detected in the nestmates' heads and the time they had groomed in different intervals before the end of the experiment (Supplementary Fig. 4b).

We tested by linear regression if the number of pellets produced per group (including only groups containing at least one pathogen-exposed individual FF, Ff, ff, FC, ff; $n = 82$ replicates, as the CC groups did not produce pellets) was dependent on group treatment, and whether the number of spores per pellet differed among treatment groups, using a Kruskal–Wallis test (Supplementary Fig. 5a). For all observed 45 pellet expulsions by nestmates representing the second to fourth expulsed pellet of the same individual, we tested if the time the ant had spent allogrooming between observed consecutive pellet

productions depended on the phase of the experiment, by use of Kruskal–Wallis testing (Supplementary Fig. 5b).

For groups in which the two treated ants received different initial spore doses (Ff, FC, fC; $n = 4915$ grooming events by 196N from 49 replicates), we tested if the nestmate preference to groom the higher-load individual could be predicted by the initial spore load difference of the two spore-treated individuals (Supplementary Fig. 7), by calculating the log odds ratio that the nestmates groom the individual with the higher initially-applied spore dose (F in Ff, F in FC, f in fC). We further calculated for all grooming choices by the nestmates performed towards the two treated ants (total 8129 grooming decisions by 328 N, out of which 5001 grooming events towards the currently higher-loaded individual by the 196 N in the 49 replicates with two spore-treated individuals FF,Ff, ff; and 3128 grooming events towards the spore-loaded individual by the 132N in the 33 replicates with one spore- and one control-treated FC,fC), the log odds ratio to groom the individual with the current higher spore load. We calculated Spearman-rank correlations to determine how the log odds ratio to groom the current higher-load individual depended on the spore load difference between the two treated individuals for all treatment groups (Fig. 6a) and separately for groups containing only one or two spore-treated individuals (Supplementary Fig. 9). We further determined the relationship between the grooming that an ant received by others recently, and its own propensity to groom others next by calculating its log odds ratio to perform allogrooming itself as a function of time since the last received grooming by others (Fig. 6b and Supplementary Fig. 10). The dependency of grooming preference on time lag was assessed by Spearman-rank correlations.

We determined the relationship between nestmate grooming preference and spore removal efficiency i.e. the sum of spores retrieved from the nestmates' heads ($n = 328$N head samples) and the pellets produced per group ($n = 77$ pellet pools), in proportion of the total spores quantified from all ants and pellets in the replicate. We tested if spore removal correlated to the degree of grooming preference towards the currently higher-load individual in the group by fitting a linear functional model for either all groups with at least one spore-treated individual (Fig. 7a; $n = 82$ replicates), or separately for groups with two spore-treated individuals (FF, Ff, ff; $n = 49$ replicates) and for groups with one spore-treated and one sham-treated individual (FC, fC; $n = 33$ replicates; Supplementary Fig. 11). As this analysis showed that groups with higher grooming preference to the higher-load individual (better choosers) had a higher spore removal efficiency we tested the a priori hypothesis in our functional knock-out experiment that groups with free choice would be able to remove more spores than groups that were prevented from choice by calculating the ratio of the spores removed in matched setups with and without choice. We assessed the significance of this ratio (test statistic) by using a bootstrap test against a null distribution where the test statistic is computed by randomly shuffling the free-choice and no-choice replicate labels.

The Supplementary Information contains a detailed description of all statistical tests in the Supplementary Notes 3, as well as all ordering details in Supplementary Table 4. The source data to our analyses are provided as source data files, the code is available under GitHub https://zenodo.org/badge/latestdoi/609868953 with the input raw data files for the code being accessible under https://research-explorer.ista.ac.at/record/12945.

### Reporting summary
Further information on research design is available in the Nature Portfolio Reporting Summary linked to this article.

### Data availability
The data generated in this study are provided as Source Data files with this paper. In addition, raw input data files for the code have been deposited in the Research Explorer database of the Institute of Science and Technology Austria under https://research-explorer.ista.ac.at/record/12945. Source data are provided with this paper.

### Code availability
On GitHub, we provide a complete code for (i) the statistical analysis of our experimental data and (ii) for the inference of the models for the spore removal behavior in the groups of ants, studied in our work, and for the stochastic simulation of these models. The first can be used to generate data tables from output files of behavioral annotation software and spore measurements, and to produce the statistical analyses and generate the plots for the experimental data analysis. The latter performs three tasks: (1) Reads and statistically analyzes the experimental input files and stores the outcome of the analysis in a form of sufficient statistics; (2) Uses the output of the statistical analysis to infer a model of a specific type; and (3) Uses the inferred rates from the previous step along with the initial segment of the experimental data to initialize and run a stochastic simulation of the inferred model. The code is available under https://zenodo.org/badge/latestdoi/609868953.

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

## Acknowledgements

We thank Mike Bidochka for the fungal strains, the ISTA Social Immunity Team for ant collection, Hanna Leitner for experimental and molecular support, Jennifer Robb and Lukas Lindorfer for microscopy, and the LabSupport Facility at ISTA for general laboratory support. We further thank Victor Mireles, Iain Couzin, Fabian Theis and the Social Immunity Team for continued feedback throughout, and Michael Sixt, Yuko Ulrich, Koos Boomsma, Erika Dawson, Megan Kutzer and Hinrich Schulenburg for comments on the manuscript. This project has received funding from the European Research Council (ERC) under the European Union's Horizon 2020 research and innovation program (Grant No. 771402; EPIDEMICSonCHIP) to SC, from the Scientific Grant Agency of the Slovak Republic (Grant No. 1/0521/20) to KB, and the Human Frontier Science Program (Grant No. RGP0065/2012) to GT.

## Author contributions

S.C., B.C.P., K.B., and G.T. conceptualized the study. Experimental data were generated by B.C.P. and A.V.G., curated by B.C.P., and analyzed by B.C.P., K.B., and G.T. Modeling was performed by K.B. with input of G.T. Figures were created by B.C.P. and K.B. The manuscript was written by S.C., G.T., B.C.P., and K.B. and approved by all authors. Funding was obtained by S.C., K.B., and G.T.

## Competing interests

The authors declare no competing interests.
