## [Peer Review File · Nature Communications]

Dynamic pathogen detection and social feedback shape collective hygiene in antsReviewers' Comments:

Reviewer #1:

Remarks to the Author:

This is an interesting and creative paper about insect social immunity. The authors used the behavioral quantification and pathogen quantification to detailedly detect the dynamic hygienical behavior (selfgrooming, allogrooming and suppressing grooming) against fungal infections. Specially, the authors focused on the important role of the social feedback in collective disease defense and described the process of social feedback during performing the sanitary behaviors, which is very instructive for the other colleagues. Additionally, this research used the probabilistic modeling to found the grooming bias of nestmates towards the higher-load individual is more pronounced when the two treated individuals clearly differ in load, which suggests that Such constant updating of pathogen load information may be key for the ants to react dynamically to changes in disease risk. Totally, this work gave the new studying methods (the combination of biology and mathematics) and ideas (the significance of social feedback) on the social immunity and social behaviors in social insects, which will be helpful for the readers. Therefore, I suggest that this article can be accepted in *Nature Communications* after minor revision.

Detailed comments:

1. Line 59-64, when the authors talked about the social immunity and sanitary behaviors, the references were mainly involved in ants and bees. Some termite-related papers should be cited here, for example:

Liu, L., Zhao, X. Y., Tang, Q. B., Lei, C. L., & Huang, Q. Y. (2019). The mechanisms of social immunity against fungal infections in eusocial insects. *Toxins*, 11(5), 244.

Liu, L., Wang, W., Liu, Y., Sun, P., Lei, C., & Huang, Q. (2019). The influence of allogrooming behavior on individual innate immunity in the subterranean termite *Reticulitermes chinensis* (Isoptera: Rhinotermitidae). *Journal of Insect Science*, 19(1), 6.

Hassan, A., Mehmood, N., Xu, H., Usman, H. M., Wu, J., Liu, L., & Huang, Q. (2022).

Phosphofructokinase downregulation disturbed termite social behaviors and immunity against fungal infections. *Entomologia Generalis*. 10.1127/entomologia/2022/1444.

2.Line 847 "with a with a" should be changed into "with a".

3.The supplementary information is detailed but is too long and too difficult to understand some parts such as mathematic modelling. Thus, hope that the authors make them more easy for reading and understanding.

Reviewer #2:

Remarks to the Author:

**** SUMMARY AND RECOMMENDATION:** The manuscript provides compelling empirical evidence of modulations in grooming behavior in ant colonies that is consistent with a social-immunity hypothesis. The manuscript also presents an impressive set of modeling tools to better understand the individual-level dynamics of these social behaviors, but a few important aspects of those models need more maturation before they are as convincing as the empirical results. Effectively, this manuscript tests mechanistic hypotheses of social immunity. However, it concludes with several strong claims about assumed benefits and costs at both the individual and group levels without any empirical justification for those benefits and costs. Speculation about ultimate causation would be fine in discussion, but the current presentation is more strongly phrased than that. In sum, I recommend this manuscript for publication after a revision which will resolve a few important questions about the theoretical models and temper some claims about functional significance.

** GENERAL ASSESSMENT

This manuscript aims to better understand the individual-level contribution to cooperative immune defense in ant colonies. To do so, the authors form small colony isolates of six ants each and then study each of the six ant's social behaviors before and after two of six ants are each infected with one of three different levels of (fluorescence-labeled) fungal pathogen (high, low, and control/none). The authors then observe how often each of the six ants are idle, self-grooming (and chemically self disinfecting), or allogrooming. At the end of the experiment, the pathogen levels present in each ant were quantified along with the number and pathogen levels in infrabuccal pellets produced by the ants. The labeling of the pathogens allowed the authors to study the total pathogen load at the end of the experiment as well as the distribution across the two possible sources of the pathogens. The major and most compelling contribution of this manuscript is the connection between changes in grooming behavior (before and after the introduction of pathogen-infected ants) with the state of the ant at the instant of the treatment. They find that all introduced ants (even control ants with no fungal load) reduced time performing grooming and increased time receiving grooming, and this pattern was reversed in those undisturbed ants that were not removed during the treatment instant. Moreover, all ants (except for the introduced control ants, which showed no significant difference) increased self-grooming after the treatment event, but infected ants also increased uptake of self-produced chemical disinfectant (whereas undisturbed ants actually decreased the use of self-produced chemical disinfectant). These patterns are consistent with: (a) infected individuals shifting their behaviors to reduce their own pathogen loads while also minimizing the transfer of pathogens to others; and (b) healthy individuals increasing allogrooming effort (while reducing their own metabolic investment in disinfectant) to remove the colony threat while also possibly also being exposed to sufficiently low levels of pathogen to receive some immunity. In other words, these observed behavioral patterns confirm individual-level mechanisms likely to provide social immunity to ant colonies. These results alone are noteworthy and deserving of publication (although I do have a couple of small questions that I will raise later).

The authors have also put significant effort into building quantitative models to act as deeper lenses on the social behavior they've observed. Some of these models are clear and compelling, but others may require a little more investigation before they are ready for dissemination. In the first category, the authors have presented an elegant model for back-computing spore load on each individual spore-treated ant at any time in the experiment based on the pathogen quantification at the end of the experiment. The authors noted that the number of spores found in each collected infrabuccal pellet was independent of treatment, but treatments with higher overall spore loads produced higher numbers of pellets. Thus, the spore-removal process could potentially be modeled with a Michaelis-Menten model of enzymatic kinetics (or, equivalently, a type-II functional response where high spore loads are rate limited). The authors validate this model by showing that it (and not other alternative models) is able to accurately back-compute the known initial spore load on treated ants. This model allowed for estimating spore load on spore-treated ants at any time in the behavioral observation, which would allow for testing whether (detectable) spore load plays a role in modulating the behavioral transitions of other ants. Although the back-computation of spore load was very compelling to me, I was less convinced by the behavioral model the authors converged around using the data from the spore decay model. The authors ultimately conclude that individual ant behavior is most consistent with each ant deciding to groom another ant with a probability that is graded with the pathogen load of the other ant. Although the graded-probability, sequential sampling model that the authors propose has merit and appears to fit data well, I was not convinced that the authors chose the best alternative/null models to contrast it with. Generally speaking, each of the presented null models felt more like a "strawman" than a true competitor, and there were some obvious (and apparently high-performing) null models that were left out of formal discussion. For example, in the text above Figure E in the supplementary behavior, the authors point out that the graded response (Eq.6) is a kind of compromise between a behavior that selects an ant for allogrooming at random and a behavior that selects only the most-infected ant. In particular, if the parameter L_0 is made to be very large, then the slope of the graded response will be so shallow that all loads will have the same probability of

receiving grooming (and thus allogrooming will be randomly given). With this in mind, the pattern shown in left panel (bars) of Figure E is very notable. In particular, when $L_0=2.5e4$, which is a value lower than the setting the authors choose to use in the main paper, the background probability parameter epsilon has a major impact on the prediction error. However, when $L_0=5e4$ (the value used by the authors) or higher, the epsilon parameter makes almost no difference, and the prediction error is flat across all L_0 values and comparable to the prediction error associated with the model chosen by the authors as the best decision-making model. If we assume that L_0 in this high range effectively flattens the graded response into a constant, we see that the small range of epsilon used by the authors corresponds to a fixed acceptance probability of between 5% and 10%. Given these results, it seems necessary for the authors to test their graded-probability sequential-sampling model against a (best-fitting) fixed-probability sequential sampling model. I believe the authors must have chosen to not do this because they were under the impression that their "random" model was already doing this. However, their "random" model is not a sequential-sampling model as it considers all ants at once and selects one of them for grooming with a uniform probability (somewhat like an "ants in an urn" model). A sequential-sampling model with a low probability of acceptance will have a very different time course. So, I would urge the authors to explicitly consider a sequential sampling model with a FIXED acceptance probability; optimize for the best probability and compare its performance to the graded probability (which the authors might want to consider relating to logistic regression, for the interpretative benefit to the more traditional life-science reader).

In the bottom third of the main manuscript, the authors discuss the significance of the functional significance of their results. Given that the manuscript is effectively chasing mechanistic questions, some of the functional language seems a bit strong. For example, on lines 199 and 200 at the end of page 8 (still in the DISCUSSION), the authors state that grooming biases in the individual ant decisions "accumulate to a measurable benefit at the collective level." Hidden in this statement is the hypothesis that the colony is better off with these behaviors than without, but the authors have not done any formal test of this particular functional hypothesis. For example, the authors have not created isolates where each of the six ants are prevented from interacting with each other (as a control). It seems very likely that the authors are correct and these behaviors are conferring social immunity, but the phrase "measurable benefit" seems to suggest the authors have quantified how much better off the ants are with the behaviors than without, and that was not what was done in this manuscript. That said, the authors do develop an "exploration--exploitation model" to formally capture the time costs of different grooming decision-making strategies so as to predict which strategy should be selected for under which situations. However, the exploration--exploitation tradeoff is more about deciding when to commit to one of a set of discrete options, which does not seem to be operating in this case. The tradeoff that seems to be more relevant for the argument the authors are making in this manuscript is the (more fundamental) speed--accuracy tradeoff. That said, it seems strange to hold the "MAX" model (where an ant deliberately probes every ant until deciding which is maximally infected) up as being "most accurate." In reality, social-immunity behaviors should be viewed in the same light as other sequential-sampling behaviors in nature -- as in optimal foraging theory (OFT). Instead of "calories per unit time" being the operational fitness proxy, "spores removed per unit time" should be used. Ants bumping into other ants, deciding to spend time cleaning them, and then moving on to search for new ants is (just like in foraging theory) a Markov renewal--reward process. Ants with different spore loads (and handling times) are analogous to prey items with different caloric gains (and handling times). Ants deciding to pass up one infected ant in order to treat another are no different than predators deciding to pass up one low-profitability prey to only spend time on certain high-profitability prey. To me, the benchmark model should be a sequential sampling model that maximizes spore treatment rate, and observed ant behaviors could be compared to hypothetical information-limited heuristics that theoretically approximate the benchmark model. This suggestion is just meant to be an example of what I would imagine a more functional investigation to look like. My larger point is that the manuscript may benefit from shifting the discussion more toward mechanism than function. I do not believe the authors can operationally define a "perfect" ant behavior at the individual level, and so it is strange to read about individual ant behaviors being called "imperfect" (line 264), and it is difficult to take for granted that the exhibited behaviors provide "such immediate

functional benefits" (line 290) without seeing how a colony behaving some other way might perform.

** SPECIFIC COMMENTS TO AUTHORS

The list below is meant to provide fine-scale feedback or suggestions to the authors at indicated lines in the manuscript.

Line 51 (Main Text): It might be helpful for a reader if the predictions of the social-immunity hypothesis/hypotheses were stated alongside the introduction to the experimental design. This may help a reader better understand choices and the gathered data. At the moment, the introduction to the experiment makes things sound more descriptive/exploratory than they actually are.

Line 56: I'm surprised that social wasps are not listed (and that "bees" is not "social bees", as there not all bees are social).

Lines 98,99: Results are referenced here where untreated nestmates increase self-grooming (but decrease use of disinfectant). I have not seen many comments throughout the manuscript about the cues (or signals) these untreated nestmates are using to release these changes in behavior. Are they simply responding to changes in grooming experience? Or could there be explicit chemical signalling indicating that a disturbed ant in the nest may have flagged itself as potentially infected? It immediately seemed peculiar to me that the undisturbed nestmates showed such a strong effect (especially in one case where the disturbed controls did not), and so I recommend commenting on this interesting result.

Line 114: I recommend adding "at the end of the experiment" (or similar) after "spore quantification". In my first pass at reading this passage (before I was more familiar with the methods), I had a hard time determining that "spore quantification" was the destructive method at the end of the experiment (as opposed to, for example, some less precise imaging technique making use of the fluorescence). So a hint/reminder that this is at the end of the experiment would be helpful.

Lines 115--118: I had a hard time understanding the comments about the correlation data within these lines. For me, this was much better explained within the caption of Figure S3.

Lines 125--126: I am not sure your message is best communicated to a reader by referencing a Michaelis--Menten model. You should certainly reference the chemical kinetics in your supplementary model, but here the salient feature is the saturation function representing spore removal rate. You could either say that it is modeled with a Hill function or even appeal to basic ecology and describe it as a type-II functional response.

Line 133: In this line, it is suggested that ants are in two-alternative forced choice trials. Without putting an animal in a Skinner box, it seems safer to assume that sequential choice is really the only appropriate model. And a graded acceptance probability would still produce apparent preferences in apparent two-choice decisions. So what about the behavioral observations really justifies saying that an ant is deciding between two ants? If you have a way to clearly identify a simultaneous choice over a couple of sequential choices, then state exactly how you identified these simultaneous choices in the data.

Line 162: If ants are really making allogrooming decisions according to a graded probability of spore load, then the number of visits between allogrooming events should be overdispersed relative to a geometric distribution. In other words, it should be better described by a beta-geometric distribution than a geometric distribution. You should be able to get the distribution (number of non-grooming encounters) between events where an ant performed grooming on another; consider looking at how geometric that distribution looks. If it can be well described by a geometric distribution (no

overdispersion), then that suggests a fixed probability. If it is overdispersed (beta-geometric), then the graded probability may have more support.

Line 164: What does "error-prone" mean in this context? Are you considering it to be an "error" if an ant passes up an infected ant (as in a type-II error/false negative)? Borrowing from some text above, if the ant is maximizing its spore removal rate over time, then passing up an ant with a low spore load may be the optimal sequential decision as it minimizes the opportunity cost. Either be more specific about what you mean by "error", or consider removing all references to error as fitness is really coming in at the colony level anyway.

Line 168: As discussed above, the sequential-sampling model with $L_0 = \infty$ (i.e., with epsilon only) is a far more important null model to test against. As discussed above, supplementary figures already suggest that a model with a very large L_0 performs as well as the best-fitting sequential-sampling model with L_0 fixed. Letting $L_0 = \infty$ turns the sequential-sampling model into one where the acceptance probability is fixed with respect to load. It is important to test whether this fixed acceptance probability performs differently than the graded probability. Note that fixing the acceptance probability is very different than choosing uniformly among all ants at every choice.

Line 169: This sentence contrasts the sequential-choice rule with an alternative where ants were choosing targets "randomly". Technically, both strategies are "random", but one of them uses a probability to decide whether to accept or ignore an encountered ant, and the other chooses an ant from a uniform distribution. Avoid using "randomly" when what is probably meant is "uniformly."

Line 200: "measurable benefit at the collective level" -- Compared to what? Do you have a no-behavior benchmark to compare to?

Line 201: I had a difficult time reading the sentence coming from line 200 on the previous page. I would recommend removing both commas (before "overall" and after "group").

Line 204: Use of "random" again. The sequential-sampling model is ALSO randomly choosing. "Choosing uniformly" seems better than "choosing at random".

Line 205: "builds up to highly efficient pathogen removal" -- relative to what benchmark? What is the control?

Line 214: It is stated that the "maximum rule" requires "remembering pathogen loads on every group member". Certainly a sequential-sampling model with a plastic deterministic threshold would only have to remember the highest load it has seen so far. If that threshold started high and decayed over time, then eventually only the highest would be serviced, but the agent would not need to remember all loads at the same time.

Line 220: "convincingly" can be a little off-putting for readers, particularly those who are not yet convinced

Line 223: Unclear that exploration--exploitation tradeoff is really the right tradeoff. Seems more like speed--accuracy tradeoff in this case.

Line 223: Rather than considering a "maximum model" where the ant has to slow down to pay a time cost to take a pathogen measurement, it may also be worthwhile to consider an ant that can take instantaneous pathogen measurements so long as it is moving sufficiently slow. The classic work of Gendron and Staddon (1983) discussed how sensory constraints can limit the speed of predators, which reduces their intake rate but also allows them to actually find prey. Reducing speed is another kind of time cost. Perhaps ants stopping to antennate each other is more like the model you have in mind (but were such session observed?), but I just wanted to point out that there are other ways in

which motion can accommodate higher sensing accuracy.

Line 225: "convincingly" again

Lines 237--240: Here, it initially states that grooming suppression occurs in all ants even in the absence of pathogenic threat", but then it concludes that "untreated nestmates remained nearly unchanged." These two statements seem to be contradictions.

Line 250: Another use of the word "random" when "uniform" is probably more accurate.

Line 287: Should "assure" be "ensure"? Or "help to ensure"?

Line 475, Figure 1e: I recommend adding color codes in a legend somewhere.

Line 475, Figure 1g: Rather than using arbitrary bins just to plot this histogram, why not present the expected CDF and the empirical CDF? That will eliminate the need for binning and will make the Kolmogorov--Smirnov distance between distributions more visual.

Line 494 and 495: "does not systematically depend on current load proportion (42/45 pairwise KS tests $p \leq 0.05$)" -- Is the $p \leq 0.05$ correct? If so, then is the "does not systematically depend" correct?

Line 503, Figure 2: As described above, would be good to plot performance of sequential sampling model with acceptance probability fixed (but optimized).

Line 533, Figure 4d: Why is there an initial difference between nestmates and pre-treatment?

Line 636: "received a different VARIANT of the two distinctively..." (word "variant" is missing)

Line 679: The mechanism underlying how the untreated nestmates detected that they should reduce their poison uptake behavior is not obvious to me. It seems like it should be commented on.

Supplementary Material, page 19: Extra space before "where y_i are the centers..."

Supplementary Material, page 23: "standard SSE algorithm" should probably be "standard SSA algorithm".

Supplementary Material, page 27: See comments above about how Figure E demonstrates that a fixed-probability sequential-sampling model would likely perform very well.

Supplementary Material, page 31: Probably more like a speed--accuracy tradeoff than an exploration--exploitation tradeoff.

Thank you for the opportunity to review this manuscript.

--Theodore (Ted) P. Pavlic

Reviewer #3:
Remarks to the Author:

Review of "Dynamic pathogen detection and social feedback shape collective hygiene in ants" by Barbara Casillas-Pérez et al. for Nature Communications

It is well known that social insects are collectively able to suppress the spread of diseases in their societies. Actually, this behavior is a requirement for their long-lived and prolific nests. Most research on the underlying mechanism has been conducted in honeybees and particularly in ants; the Cremer group is at the fore-front of this research. Here again they bring the research field to a new level, by combining individual observations on grooming and its effectiveness with simulations of individual decisions that could explain the collective disease management patterns in the studied ant species (and probably also other eusocial insects). Behavioral analyses revealed that ants in groups of six increase grooming and preferentially target highly-infectious conspecifics when perceiving high pathogen load, but reduce grooming after having been groomed by others. Individual decisions rely only on local and incomplete information, but simulations suggest that such decision rules can lead to efficient collective disease defense of whole nests.

Overall both the new methods and the simulations applied will bring the field a major step forward and will inspire many other researchers working with this and other model systems. The article is perfectly written, very well illustrated and obviously went through several rounds of expert review (also seen in the acknowledgements section). I find methods and results very well explained and introduced, and all claims and conclusions fully supported. The findings are very exciting and Nature Communications is a great fit for this important article.

During reading the article I came up with several comments that were all finally resolved in the supplementary methods section. However, two of these things I suggest to briefly mention already in the main manuscript instead of the supplements: (1) obviously it is very important that observers are blind to the manipulations of individuals, so it is great that observers were blind to the color coding – please mention that in the main text already; (2) it is great that you quantified spore loads on individuals after applying your treatments, but this is not mentioned in the main text – please change that. Furthermore, I found Fig 1a misleading in this respect. This figure looks like a result of your study (it is presented together with other results), but is actually a description of the setup. However, you have quantified spore loads, so instead of using just dummy numbers, please use the actual numbers with variances you give in the supplements.

So apart from these two minor comments I have no further suggestions to change before acceptance for publication. However, I just would like to point out that I am an experimental entomologist and do not feel competent to evaluate the simulations part.

POINT-BY-POINT RESPONSE TO REVIEWER COMMENTS

Reviewer #1 (Remarks to the Author):

This is an interesting and creative paper about insect social immunity. The authors used the behavioral quantification and pathogen quantification to detailedly detect the dynamic hygienical behavior (selfgrooming, allogrooming and suppressing grooming) against fungal infections. Specially, the authors focused on the important role of the social feedback in collective disease defense and described the process of social feedback during performing the sanitary behaviors, which is very instructive for the other colleagues. Additionally, this research used the probabilistic modeling to found the grooming bias of nestmates towards the higher-load individual is more pronounced when the two treated individuals clearly differ in load, which suggests that Such constant updating of pathogen load information may be key for the ants to react dynamically to changes in disease risk. Totally, this work gave the new studying methods (the combination of biology and mathematics) and ideas (the significance of social feedback) on the social immunity and social behaviors in social insects, which will be helpful for the readers. Therefore, I suggest that this article can be accepted in *Nature Communications* after minor revision.

Thank you for a positive evaluation of our work.

Detailed comments:

1. Line 59-64, when the authors talked about the social immunity and sanitary behaviors, the references were mainly involved in ants and bees. Some termite-related papers should be cited here, for example:

Liu, L., Zhao, X. Y., Tang, Q. B., Lei, C. L., & Huang, Q. Y. (2019). The mechanisms of social immunity against fungal infections in eusocial insects. *Toxins*, 11(5), 244.

Liu, L., Wang, W., Liu, Y., Sun, P., Lei, C., & Huang, Q. (2019). The influence of allogrooming behavior on individual innate immunity in the subterranean termite *Reticulitermes chinensis* (Isoptera: Rhinotermitidae). *Journal of Insect Science*, 19(1), 6.

Hassan, A., Mehmood, N., Xu, H., Usman, H. M., Wu, J., Liu, L., & Huang, Q. (2022).

Phosphofructokinase downregulation disturbed termite social behaviors and immunity against fungal infections. *Entomologia Generalis*. 10.1127/entomologia/2022/1444.

Thank you for pointing out the missing emphasis on termites. We agree that our citations were biased to the social Hymenoptera and now added three references on termites (refs 25,26 and 31).

2.Line 847 "with a with a" should be changed into "with a".

We removed the duplication.

3.The supplementary information is detailed but is too long and too difficult to understand some parts such as mathematic modelling. Thus, hope that the authors make them more easy for reading and understanding.

We have restructured the main manuscript by including a methods section that should be comprehensible for most of the readers and that contains the general experimental and data analysis approaches. We have also restructured the Supplement to provide a more integrated section on data analysis including statistics and modeling.

For modelling and simulations, which constitute an essential part of our paper and which we kept very short in the main manuscript, we believe – in particular in light of comments by Reviewer 2 – that it is important to provide extensive detail in the supplement, so that the specialist readers that focus on those aspects can also follow our manuscript and replicate our simulations and analyses.

Taken together, we hope that the revised structure (methods in the main manuscript, restructured SI) provides essential information supporting our results that can be navigated by scientists with different backgrounds.

Reviewer #2 (Remarks to the Author):

**** SUMMARY AND RECOMMENDATION:** The manuscript provides compelling empirical evidence of modulations in grooming behavior in ant colonies that is consistent with a social-immunity hypothesis. The manuscript also presents an impressive set of modeling tools to better understand the individual-level dynamics of these social behaviors, but a few important aspects of those models need more maturation before they are as convincing as the empirical results. Effectively, this manuscript tests mechanistic hypotheses of social immunity. However, it concludes with several strong claims about assumed benefits and costs at both the individual and group levels without any empirical justification for those benefits and costs. Speculation about ultimate causation would be fine in discussion, but the current presentation is more strongly phrased than that. In sum, I recommend this manuscript for publication after a revision which will resolve a few important questions about the theoretical models and temper some claims about functional significance. Thank you very much for this evaluation and the in-depth review, highlighting where and how to improve our work. As detailed below, we have followed all suggestions, by comparing our inferred model to a greater variety of (no-strawman) null models, improving our modeling for the exploration-exploitation trade-off, performing additional behavioral analysis on the sequences of non-grooming events (antennations) preceding grooming decisions, as well as providing new experimental evidence for the functional benefits of decision making, by comparing the free-choice situation with experimental conditions preventing choice. We think that these additions have greatly improved the conclusions we can draw. Thank you for all your thoughtful suggestions.

**** GENERAL ASSESSMENT**

This manuscript aims to better understand the individual-level contribution to cooperative immune defense in ant colonies. To do so, the authors form small colony isolates of six ants each and then study each of the six ant's social behaviors before and after two of six ants are each infected with one of three different levels of (fluorescence-labeled) fungal pathogen (high, low, and control/none). The authors then observe how often each of the six ants are idle, self-grooming (and chemically self disinfecting), or allogrooming. At the end of the experiment, the pathogen levels present in each ant were quantified along with the number and pathogen levels in infrabuccal pellets produced by the ants. The labeling of the pathogens allowed the authors to study the total pathogen load at the end of the experiment as well as the distribution across the two possible sources of the pathogens. The major and most compelling contribution of this manuscript is the connection between changes in grooming behavior (before and after the introduction of pathogen-infected ants) with the state of the ant at the instant of the treatment. They find that all introduced ants (even control ants with no fungal load) reduced time performing grooming and increased time receiving grooming, and this pattern was reversed in those undisturbed ants that were not removed during the treatment instant. Moreover, all ants (except for the introduced control ants, which showed no significant difference) increased self-grooming after the treatment event, but infected ants also increased uptake of self-produced chemical disinfectant (whereas undisturbed ants actually decreased the use of self-produced chemical disinfectant). These patterns are consistent with: (a) infected individuals shifting their behaviors to reduce their own pathogen loads while also minimizing the transfer of pathogens to others; and (b) healthy individuals increasing allogrooming effort (while reducing their own metabolic investment in disinfectant) to remove the colony threat while also possibly also being exposed to sufficiently low levels of pathogen to receive some immunity. In other words, these

observed behavioral patterns confirm individual-level mechanisms likely to provide social immunity to ant colonies. These results alone are noteworthy and deserving of publication (although I do have a couple of small questions that I will raise later).

The authors have also put significant effort into building quantitative models to act as deeper lenses on the social behavior they've observed. Some of these models are clear and compelling, but others may require a little more investigation before they are ready for dissemination. In the first category, the authors have presented an elegant model for back-computing spore load on each individual spore-treated ant at any time in the experiment based on the pathogen quantification at the end of the experiment. The authors noted that the number of spores found in each collected infrabuccal pellet was independent of treatment, but treatments with higher overall spore loads produced higher numbers of pellets. Thus, the spore-removal process could potentially be modeled with a Michaelis–Menten model of enzymatic kinetics (or, equivalently, a type-II functional response where high spore loads are rate limited). The authors validate this model by showing that it (and not other alternative models) is able to accurately back-compute the known initial spore load on treated ants. This model allowed for estimating spore load on spore-treated ants at any time in the behavioral observation, which would allow for testing whether (detectable) spore load plays a role in modulating the behavioral transitions of other ants. Although the back-computation of spore load was very compelling to me, I was less convinced by the behavioral model the authors converged around using the data from the spore decay model. The authors ultimately conclude that individual ant behavior is most consistent with each ant deciding to groom another ant with a probability that is graded with the pathogen load of the other ant. Although the graded-probability, sequential sampling model that the authors propose has merit and appears to fit data well, I was not convinced that the authors chose the best alternative/null models to contrast it with. Generally speaking, each of the presented null models felt more like a "strawman" than a true competitor, and there were some obvious (and apparently high-performing) null models that were left out of formal discussion. For example, in the text above Figure E in the supplementary behavior, the authors point out that the graded response (Eq.6) is a kind of compromise between a behavior that selects an ant for allogrooming at random and a behavior that selects only the most-infected ant. In particular, if the parameter L_0 is made to be very large, then the slope of the graded response will be so shallow that all loads will have the same probability of receiving grooming (and thus allogrooming will be randomly given). With this in mind, the pattern shown in left panel (bars) of Figure E is very notable. In particular, when $L_0=2.5e4$, which is a value lower than the setting the authors choose to use in the main paper, the background probability parameter ϵ has a major impact on the prediction error. However, when $L_0=5e4$ (the value used by the authors) or higher, the ϵ parameter makes almost no difference, and the prediction error is flat across all L_0 values and comparable to the prediction error associated with the model chosen by the authors as the best decision-making model. If we assume that L_0 in this high range effectively flattens the graded response into a constant, we see that the small range of ϵ used by the authors corresponds to a fixed acceptance probability of between 5% and 10%. Given these results, it seems necessary for the authors to test their graded-probability sequential-sampling model against a (best-fitting) fixed-probability sequential sampling model. I believe the authors must have chosen to not do this because they were under the impression that their "random" model was already doing this. However, their "random" model is not a sequential-sampling model as it considers all ants at once and selects one of them for grooming with a uniform probability (somewhat like an "ants in an urn" model). A sequential-sampling model with a low probability of acceptance will have a very different time course. So, I would urge the authors to explicitly consider a sequential sampling model with a FIXED acceptance probability; optimize for the best probability and compare its performance to the graded probability (which the authors might want to consider relating to logistic regression, for the interpretative benefit to the more traditional life-science reader).

Thank you for the detailed comment. We agree that the analysis of the decision rule by which ants choose the allogrooming targets could have been strengthened. Consequently, we have followed your recommendations by evaluating the rule you suggested above (optimal fixed probabilities for selecting an allogrooming target), as well as other rules which we considered to be either illustrative or plausible. We report on this in the revised Fig. 3d and the associated discussion. Specifically, we now consider the original “random rule” (which for clarity we renamed into “uniform random” rule) as a “strawman” null model. This is to be compared to the “maximum rule” model that we had in the initial submission – the rule that assumes global knowledge of the load on all ants.

The uniform random rule and the maximum rule can now also be compared against two new and more sophisticated alternatives that do not assume dynamic updating of the spore load information. The first non-dynamic alternative is what we call ‘sequential rule without updating’. This is equivalent to the data-favored sequential rule with dynamic load updating in its mathematical form and its parameters, with the sole distinction that the loads upon which the choice is made are the initial spore loads that never change with time. The second non-dynamic alternative is based on the ‘optimal probabilities’ idea suggested specifically by the referee. These new alternatives are described in the manuscript (lines 209-232) and detailed in Supplementary information pages 20, 35-36. We have also updated our model figure (new Fig 3d) to include these new non-trivial alternatives.

In sum, our sequential choice based on updated load information still provides the best match to data even including new (no-strawman) alternatives. We write in the manuscript: *“Finally, our analysis revealed that a preference of an ant to groom the higher-load individual (Fig. 2c) is most parsimoniously explained by what we call a ‘sequential-choice rule’: an ant probes one group member after the other, and commits to groom the currently probed ant with a probability that increases with the probed ant’s current spore load (Fig. 3d; Supplementary Model). Notably, this choice rule does not require an ant to compare (and remember) the spore load on multiple other ants. While the empirically identified rule may not be optimal in terms of the total spore removal efficiency as a consequence of its local and sequential nature, it nevertheless leads to a systematic bias towards preferentially ‘choosing’ higher-load individuals; similar collective choices that do not require individual comparisons (and thus global knowledge) have been reported in other contexts, like nest choice in ants^{14,18,43}. Simulations using the sequential-choice rule matched data better than various alternatives (Fig. 3d). The most naïve alternative we explored was one in which ants chose their grooming targets uniformly at random; this alternative, however, provided the worst match to data. Next, we explored two less trivial alternatives in which ants choose their grooming targets based on information that was not dynamically updated. In the first no-updating alternative, ants did perform sequential choice, but using initial spore loads only; removing dynamic load-updating thus substantially worsened fit to data. In the second no-updating alternative, ants used fixed (but non-uniform) probabilities to choose amongst their F, f, C, N targets. Even when these probabilities themselves were fitted to maximize agreement with data, this alternative still underperformed sequential choice. The last alternative we considered allowed for dynamic updating and assumed that ants could acquire complete information about all current loads, to deterministically pick the ant with highest current load to groom (‘maximum rule’). While such a hypothetical rule would result in the most rapid removal of spores from the colony, it was also not supported by data.”*

In the bottom third of the main manuscript, the authors discuss the significance of the functional significance of their results. Given that the manuscript is effectively chasing mechanistic questions, some of the functional language seems a bit strong. For example, on lines 199 and 200 at the end of page 8 (still in the DISCUSSION), the authors state that grooming biases in the individual ant decisions “accumulate to a measurable benefit at the collective level.” Hidden in this statement is the hypothesis that the colony is better off with these behaviors than without, but the authors have not done any formal test of this particular functional hypothesis. For example, the authors have not created isolates where each of the six ants are prevented from interacting with each other (as a control). It seems very likely that the authors are correct and these behaviors are conferring social

immunity, but the phrase "measurable benefit" seems to suggest the authors have quantified how much better off the ants are with the behaviors than without, and that was not what was done in this manuscript.

We agree that our initial manuscript only provided correlative evidence that groups are able to remove more spores, if they are able to make correct grooming decisions more frequently (old Fig. 4b).

We have now performed a new "functional knock-out" experiment in which we tested for the hypothesis derived from the main experiment that choice is beneficial for total spore removal. Thus, preventing choice should lead to reduced total spore removal by the ants. In this additional experiment, we permitted certain ant groups a free choice between two treated ants, whereas we prevented such a choice in other groups. We then compared the spore removal efficiency in the groups with and without choice, while controlling for the composition and total spore load in both cases.

In detail, we set up groups of four nestmates and two treated individuals (just as in our main experiment), and contrasted spore removal in this setup to a new benchmark "without choice". This benchmark consisted of two "half-groups": each original group of 6 ants (two treated plus four nestmates) was split into two halves, each consisting of one treated ant and two nestmates. We focused in our experiment on the period of highest grooming activity, when most grooming events (and hence choices) occur (i.e. the first 20 min after exposure, see Fig. 4 timeline). We then compared the TOTAL removed spores (spores in the heads of nestmates and expelled as pellets) in groups with choice (6 ants, 2 of which treated) and without choice (two corresponding groups each with 1 treated and 2 nestmates).

To this end, we set up different choice situations, where one ant was always treated with a high dose (1×10^9 spores/ml), and the second either with the same dose (difficult choice), half or quarter of the dose (moderate choice difficulty), or the spore-free control treatment (easy choice). We again quantified the spore removal by ddPCR and determined the ratio of spore removal with / without choice, i.e. spores removed by the complete groups / spores removed by the two respective halves (e.g. $4N+FF / 2 \cdot (2N+F)$, or $4N+Ff / (2N+F)+(2N+f)$). Very consistently, we observed ~1.5-fold higher removal rate in choice groups compared to matched no-choice groups. In aggregate, this difference was statistically significant, as we demonstrated by testing against a null distribution constructed specifically for this test using bootstrapping.

This experiment confirmed and extended our previous results of higher spore removal by 'better choosers', as we found that situations allowing choice led to higher spore removal compared to situations in which choice was experimentally prevented. While it is challenging to experimentally "block choice" while ensuring the absence of any potential behaviorally-relevant side effects of this manipulation, we believe these new experimental results, motivated by the referee's query, are the closest we can get to provide not only correlational, but causative evidence for the importance of choice.

We include the new experiment following the old correlational evidence in lines 342-252 (and methods section lines 526-548) and show the results in a new figure (Fig. 7b). We therefore also feel confident to report on these combined data as providing evidence for a functional benefit of informed choice for collective spore removal.

That said, the authors do develop an "exploration--exploitation model" to formally capture the time costs of different grooming decision-making strategies so as to predict which strategy should be selected for under which situations. However, the exploration--exploitation tradeoff is more about deciding when to commit to one of a set of discrete options, which does not seem to be operating in this case. The tradeoff that seems to be more relevant for the argument the authors are making in this manuscript is the (more fundamental) speed--accuracy tradeoff.

Thank you for this comment. We carefully considered the terminology and framework of speed-accuracy tradeoff, but believe that the formulation of exploration-exploitation would be more easily

understandable for the broader readership of *Nature Communications*. In neuroscience, for instance, speed-accuracy often applies to a choice (e.g., in sensory-motor integration) that can be done more accurately if the integration of sensory evidence can take a longer time: ultimately, the same qualitative choice / action is taken, only at a higher level of “quantitative precision” (e.g., hitting the target with one’s hand when one has had enough time for sensory integration to extrapolate its future motion, vs missing the target because the time for evidence accumulation was limited). In exploration-exploitation, different targets (e.g., pastures for foraging, or here, ants of differing load to be groomed) can be probed but only at a time-cost which must be traded off against the performance on the found target (e.g., feeding on the pasture where the animals have committed to stay, or grooming an ant that is chosen).

While we agree with the referee that social immunity problem that we study can also be framed in terms of a speed-accuracy tradeoff, we find exploration-exploitation framework to also be suitable, and since this is the framework that guided our thinking originally we are inclined to keep the same formulation in our revised manuscript.

Should you believe, however, that speed-accuracy substantially better captures the nature of the tradeoff, we are willing to make this change in terminology – which is a minor change to the manuscript. In this case, we would be happy to receive a more guided suggestion on how to include also the examples of optimal foraging theory invoked below.

That said, it seems strange to hold the "MAX" model (where an ant deliberately probes every ant until deciding which is maximally infected) up as being "most accurate." In reality, social-immunity behaviors should be viewed in the same light as other sequential-sampling behaviors in nature -- as in optimal foraging theory (OFT). Instead of "calories per unit time" being the operational fitness proxy, "spores removed per unit time" should be used. Ants bumping into other ants, deciding to spend time cleaning them, and then moving on to search for new ants is (just like in foraging theory) a Markov renewal--reward process. Ants with different spore loads (and handling times) are analogous to prey items with different caloric gains (and handling times). Ants deciding to pass up one infected ant in order to treat another are no different than predators deciding to pass up one low-profitability prey to only spend time on certain high-profitability prey.

Thank you for drawing this analogy. In the literature that we are familiar with, problems in foraging are also analyzed using the exploration-exploitation framework, which is in line with our understanding that the two concepts are often used interchangeably. As stated above, we are happy to make the terminology change if considered beneficial to the readers.

To me, the benchmark model should be a sequential sampling model that maximizes spore treatment rate, and observed ant behaviors could be compared to hypothetical information-limited heuristics that theoretically approximate the benchmark model. This suggestion is just meant to be an example of what I would imagine a more functional investigation to look like. My larger point is that the manuscript may benefit from shifting the discussion more toward mechanism than function. We made progress in the suggested direction – towards linking the mechanism of decision making to function – but substantial further work including different experimental design would be necessary to really pinpoint one specific mechanism to the exclusion of others. This is currently beyond the scope of what we can deliver.

Nevertheless, while our work is rooted in functional comparisons (e.g., the maximum-rule model of “how much spores the ants *could* remove assuming they were making an optimal choice to groom the currently highest-loaded ant without any constraints and with perfect global information” vs the uniform-random rule model of “the grooming target is always picked uniformly at random”) we have made the following modifications in line with the reviewer’s suggestion:

(i) We have considered new, non-strawman decision rules for grooming and tested their compatibility with data, as suggested by the referee (main text, starting line 209; also new Fig. 3d);

(ii) We found further experimental support for the mechanism that could underlie sequential choice in terms of transient antennation events (main text, starting line 234; also new Supplementary Figure 7);

(iii) We have explored the space of optimal exploration-exploitation models that would permit efficient colony-wide spore removal under time constraint. This new analysis highlights common features of decision rules that would well navigate the tradeoff, particularly the importance of not committing grooming-time to non-infectious nestmates. Several subtler points are also reported in the supplement (main text, starting line 280, new supplementary model text).

After making the above changes, we hope that our revised manuscript better ties together the functional outcomes that we can observe with potential mechanisms.

I do not believe the authors can operationally define a "perfect" ant behavior at the individual level, and so it is strange to read about individual ant behaviors being called "imperfect" (line 264), and it is difficult to take for granted that the exhibited behaviors provide "such immediate functional benefits" (line 290) without seeing how a colony behaving some other way might perform.

As in several of our other replies below, we agree with the referee that the terms "error-prone" or "imperfect" only make sense when defined relative to a well-defined standard (whether that standard is biologically plausible or not). We apologize for not having been clear on this point.

Our intention for using the term "imperfect" was to refer to ants not making the "optimal choice", i.e., not picking the individual for grooming with the highest current load, assuming that they could have done that in absence of any biological plausibility constraints. We have clarified (see further replies below) the wording where such situations arose.

In addition, we carried out an entirely new "knock-out experiment" (main manuscript, starting line 336) to provide direct experimental support for the functional benefit of making an informed choice about allogrooming targets. Furthermore, we have replaced "imperfect" by "noisy" in the abstract and extended our reasoning in the discussion (lines 368-370) to: *"Despite these limitations that exert their effect at the level of individual ants, the identified rules interact synergistically and amplify into efficient pathogen removal with expected epidemiological benefits at the colony level."*

** SPECIFIC COMMENTS TO AUTHORS

The list below is meant to provide fine-scale feedback or suggestions to the authors at indicated lines in the manuscript.

Line 51 (Main Text): It might be helpful for a reader if the predictions of the social-immunity hypothesis/hypotheses were stated alongside the introduction to the experimental design. This may help a reader better understand choices and the gathered data. At the moment, the introduction to the experiment makes things sound more descriptive/exploratory than they actually are.

We have added a completely new paragraph to the introduction that frames the main question which we subsequently pursue in the experiment (lines 68-85). We open this paragraph by *"To understand the individual decision-making process that forms the basis of emergent collective hygiene, we put ants into an experimental situation where they could choose how to distribute their sanitary care between two group members carrying different loads of an infectious fungal pathogen. ..."* We have also rewritten the experimental description (lines 98-118), to make it less descriptive and more integrated into the larger framework. In particular, we added (lines 99-1019: *"The untreated four nestmates therefore either faced a clear (FC, fC), less distinct (Ff) or no (FF, ff, CC) initial spore load difference between the two treated individuals"*.

Line 56: I'm surprised that social wasps are not listed (and that "bees" is not "social bees", as there not all bees are social).

We changed to “Social insects like the social bees and wasps, the ants and termites...” (line 56).

Lines 98,99: Results are referenced here where untreated nestmates increase self-grooming (but decrease use of disinfectant). I have not seen many comments throughout the manuscript about the cues (or signals) these untreated nestmates are using to release these changes in behavior. Are they simply responding to changes in grooming experience? Or could there be explicit chemical signaling indicating that a disturbed ant in the nest may have flagged itself as potentially infected? It immediately seemed peculiar to me that the undisturbed nestmates showed such a strong effect (especially in one case where the disturbed controls did not), and so I recommend commenting on this interesting result.

Thank you for suggesting an expansion of our findings, which – even if predicted by theory to reduce disease spread (Theis et 2015 Phil Trans R Soc) – to our knowledge, have not yet been reported in the literature.

To address this comment on a structural level, we have modified the manuscript by introducing a new section header (line 120: “*Pathogen exposure triggers individual and collective hygiene*”) and expanding on the corresponding content. We further split the original Fig. 1 into two parts, the first of which now exclusively reports on the changes in the individual and collective hygiene upon pathogen exposure (new Fig. 1b-d).

To address this comment on the level of new analyses, we focused on our data to establish how selfgrooming of nestmates interacts with previous allogrooming of infectious (vs non-infectious) individuals. This analysis revealed that nestmates spend more time grooming themselves after having groomed a spore-treated individual compared to having groomed another non-infectious nestmate. Interestingly, they also increase selfgrooming after having groomed a control-treated individual, yet only if the latter was part of a group with a spore-treated, but not with another control-treated individual. This reveals that not only the treatment of the partner, but also the pathogen-context of the entire group, appear to be relevant for the nestmates. Having been groomed by others, however, did not modify a nestmate’s selfgrooming, revealing that the increase in self-hygiene is not just a response to contact to a spore-treated individual, but only follows active sanitary care provisioning. We have added a new Supplementary Figure 3 and have expanded the manuscript text (see below).

We so far do not have any indication that at this early disease stage, where ants are only contaminated, but not yet infected, the ants would flag themselves. However, recent work in termites (Chen et al 2022, Insect Science) and by our group on ants (Stock et al 2023, Nature Ecology & Evolution) revealed that allogrooming of *Metarhizium*-exposed colony members can be triggered by chemical pathogen cues, like ergosterol – information that we have now added to the manuscript. We also consider it an interesting finding that the nestmates reduce their poison uptake, yet we can only speculate that this may be in response to the upregulation of poison use by the treated ants, which they could sense due to the release of the volatile formic acid (see also comment below), coupled to the fact that they themselves typically only get cross-contaminated with a low dose of pathogen.

This section now reads: lines 128-149: “...*The nestmates, on the other hand, provided high levels of sanitary caregiving, which they directed mostly towards pathogen-treated individuals, and less frequently also towards control-treated individuals. This is consistent with the observation that sanitary care in ants and other social insects like termites is triggered by chemical pathogen cues detected on the exposed individuals, such as the fungal membrane compound ergosterol^{38,39}.*

The untreated nestmates also increased selfgrooming as a response to pathogen exposure of the group (Fig. 1b). In particular, they spent more time grooming themselves after grooming a spore-loaded (F,f) individual than after grooming another untreated nestmate (N), whereas grooming a control-treated ant elicited a context-dependent response (Supplementary Fig. 3a). Notably, nestmate selfgrooming was triggered only by performing grooming towards – but not by receiving grooming from (Supplementary Fig. 3b) – infectious ants, and was therefore not a simple reaction to contact. In contrast to treated individuals who showed an increase in the use of their sanitizing poison, the

nestmates decreased the utilization of their poison, which is costly to produce⁴⁰ (Supplementary Fig. 2, Supplementary Table 2). It is still unclear whether nestmates refrain from using own poison when sensing the increased application of the volatile formic acid-rich disinfectant by the treated individuals, or whether they can assess their low risk of getting cross-contaminated with high, disease-causing pathogen levels⁴¹. Independently of the underlying mechanism, such increased self-hygiene by nestmates of infectious individuals is predicted by epidemiological modeling to evolve, as it reduces disease risk for the colony³³.”

Line 114: I recommend adding "at the end of the experiment" (or similar) after "spore quantification". In my first pass at reading this passage (before I was more familiar with the methods), I had a hard time determining that "spore quantification" was the destructive method at the end of the experiment (as opposed to, for example, some less precise imaging technique making use of the fluorescence). So a hint/reminder that this is at the end of the experiment would be helpful.

We have changed the sentences (see below), and already made clearer when describing the experimental design that the ants were sampled for spore determination after the end of the experiment (line 111-112: *"After the end of the experiment, we sampled each ant and determined the number and origin of spores separately for its head and body..."*) to emphasize the timing and destructive aspect of our method.

Lines 115--118: I had a hard time understanding the comments about the correlation data within these lines. For me, this was much better explained within the caption of Figure S3.

We have changed the text to make it easier to read (lines 155-161): *"The number and type of spores retrieved from each ant's infrabuccal pocket at the end of the experiment (as quantified by fluorescence-specific ddPCR) correlated well with the ant's grooming activity in the last sixty minutes of the experiment (min 30-90 after exposure). Earlier grooming events (min 1-30), however, were less well reflected in the stored spores, likely because the ants had already expelled these spores as a pellet in the meantime (Fig. 2a, Supplementary Fig. 4)."*

Lines 125--126: I am not sure your message is best communicated to a reader by referencing a Michaelis--Menten model. You should certainly reference the chemical kinetics in your supplementary model, but here the salient feature is the saturation function representing spore removal rate. You could either say that it is modeled with a Hill function or even appeal to basic ecology and describe it as a type-II functional response.

We now refer to our spore decay model as Type II functional response model (main text, supplement, Supplementary table) and introduce it in lines 167-171 by: *"This observation was quantitatively captured by a Type II functional response model⁴² for grooming efficiency (mathematically equivalent to a Michaelis-Menten reaction kinetics; Supplementary Table 3, Supplementary Methods), in which the spore removal rate during grooming is at saturation when the pathogen load on the infectious ant is high, but decreases at lower pathogen loads."*

Line 133: In this line, it is suggested that ants are in two-alternative forced choice trials. Without putting an animal in a Skinner box, it seems safer to assume that sequential choice is really the only appropriate model. And a graded acceptance probability would still produce apparent preferences in apparent two-choice decisions. So what about the behavioral observations really justifies saying that an ant is deciding between two ants? If you have a way to clearly identify a simultaneous choice over a couple of sequential choices, then state exactly how you identified these simultaneous choices in the data.

We apologize for being unclear. What the current line 176 (former line 133) states is only what we can actually observe empirically: i.e., that we collected all allogrooming events in which an ant targeted one of the two treated individuals, and further analyzed the probability that the ant chose the higher-loaded vs lesser-loaded individual. The ant could choose to groom other individuals (e.g.,

nestmates), but these events are not analyzed in Fig 2c and 2d. What this analyses show is the “functional outcome” of some (at that stage unknown!) process by which an ant chooses one individual over the other to groom in a well-defined subset of allogrooming events. Importantly, the mechanism behind this process could be either sequential or not, and could still produce exactly the same empirical signatures as reported in Fig 2c and 2d. We do not read the text in line 176 as favoring any particular mechanism, sequential or not, as this is indeed still an open question at that point in the manuscript.

Specifically, the observed bias in grooming could be a result of a simultaneous decision between two (or more) individuals with a graded probability that is computed based on current, dynamically updated loads (the fact that the choice needs to be based on dynamically updated loads and not static information is strongly supported in the revised manuscript, as we argue in other replies above). Alternatively, it could be a series of sequential decisions with a particular acceptance function, which will lead to an indistinguishable functional outcome. This second sequential alternative (“sequential rule with dynamic updating”), which we analyze further and simulate as part of Fig. 3d and Fig. 4, has the advantage that it is local and thus plausible, as we argue starting in line 214. We also provide new support for the plausibility of sequential choice in terms of the observation of a sequence of transient antennation events (see reply below, and text starting with line 234).

In sum, we hope that the revised manuscript makes clear our reasoning: first, our empirical report that ants are biasing their grooming towards higher-loaded individual, which is a functional outcome that could be implemented by different mechanisms; second, a proposal that a plausible mechanism is sequential choice-based on dynamically-updated load information, which fits the data and links to antennation. Our current experiments, however, do not permit us to exclude all other possible rules that could, putatively, also match the data (even though several informed alternatives that we tried in Fig 3d indeed do not match the data as well).

Line 162: If ants are really making allogrooming decisions according to a graded probability of spore load, then the number of visits between allogrooming events should be overdispersed relative to a geometric distribution. In other words, it should be better described by a beta-geometric distribution than a geometric distribution. You should be able to get the distribution (number of non-grooming encounters) between events where an ant performed grooming on another; consider looking at how geometric that distribution looks. If it can be well described by a geometric distribution (no overdispersion), then that suggests a fixed probability. If it is overdispersed (beta-geometric), then the graded probability may have more support.

Thank you for suggesting us to analyze non-grooming encounters, to find support (or not) in the ants’ behavioral sequences for either a simultaneous or sequential assessment of spore loads. We have therefore analyzed in depth the first 30 min after exposure (as most grooming occurs before minutes 20-30 after exposure) for one replicate of Ff treatment and one replicate of FC treatment. This has been most productive, as detailed below.

We find that the ants typically start a grooming event by antennation, but that there are also several transient antennation events (often directed against different individuals) before an ant makes the decision to allogroom. Antennation is a common ant behavior which the ants use for detection of chemical compounds (the antennae are their most important sensory organ with many olfactory receptors), allowing them to e.g. assess colony identity of the other individual (nestmate recognition) etc. We consider it likely that antennation can also be used in the context of spore load detection (which has been linked to the detection of particular microbial compounds, detailed above). We show these data in a new Supplementary Fig. 7, revealing that the ants’ behavior could be consistent with the postulated sequential choice rule.

Unfortunately, our subsequent (manual) analysis was limited to 305 occasions of antennation in the single Ff replicate and 225 occasions in the single FC replicate, and we do not feel confident that this is sufficiently reliable for testing the (over)dispersion that the referee suggests, especially taking into

account the variability across replicates. We agree, however, that this is a productive direction for future research.

Despite this lack of statistical evidence that we could present, the analysis suggested by the referee provided a very compelling hypothesis for the mechanism underlying sequential choice backed by our in-depth analysis of antennation. We have correspondingly amended the manuscript to write (lines 234-248): *Predictive performance of various rules for grooming choice (Fig. 3d) clearly identified the importance of continuous information updating and suggested sequential choice as a biologically-plausible procedure by which such information could be utilized by individual ants. Sequential choice predicts a clear experimental signature: the existence of multiple transient ant-ant interactions that precede, but do not immediately lead to, allogrooming. We therefore analyzed in detail the ants' behavior in two of our experimental replicates (Supplementary Fig. 7) to see if the ants may sequentially 'probe' their targets before committing to groom, preferentially, the higher-load ant. Indeed, we found bouts of antennation – a common recognition and discrimination behavior in ants⁴⁴ – preceding most allogrooming events, whereby an ant would make several transient contacts with different target individuals before finally choosing an ant to groom; typically, the chosen ant was also antennated immediately prior to grooming. While our experiments do not permit us to unambiguously and causally identify antennation as the 'probing' mechanism underpinning the sequential choice, they provide a possible mechanistic basis and correlational evidence in support of this idea.*

Line 164: What does "error-prone" mean in this context? Are you considering it to be an "error" if an ant passes up an infected ant (as in a type-II error/false negative)? Borrowing from some text above, if the ant is maximizing its spore removal rate over time, then passing up an ant with a low spore load may be the optimal sequential decision as it minimizes the opportunity cost. Either be more specific about what you mean by "error", or consider removing all references to error as fitness is really coming in at the colony level anyway.

We agree with this, since one can only talk about "errors" when they can (and should) plausibly be avoided, which requires an explanation of the context / constraints in which a biological mechanism operates. We softened our claim as follows (lines 214-217): *"While the empirically identified rule may not be optimal in terms of the total spore removal efficiency as a consequence of its local and sequential nature, it nevertheless leads to a systematic bias towards preferentially 'choosing' higher-load individuals..."* We believe that this explanation is appropriate, as we talk at the end of the same paragraph about the "maximal rule" that would have been optimal if nestmates faced no cost of any kind (time cost or otherwise) to collecting global information and always (correctly) groomed the ant with the maximal load. Such a rule may not be biologically realistic, but it sets a clear upper bound.

We hope the new wording that does not mention "error-prone" decision making is more appropriate.

Line 168: As discussed above, the sequential-sampling model with $L_0 = \infty$ (i.e., with epsilon only) is a far more important null model to test against. As discussed above, supplementary figures already suggest that a model with a very large L_0 performs as well as the best-fitting sequential-sampling model with L_0 fixed. Letting $L_0 = \infty$ turns the sequential-sampling model into one where the acceptance probability is fixed with respect to load. It is important to test whether this fixed acceptance probability performs differently than the graded probability. Note that fixing the acceptance probability is very different than choosing uniformly among all ants at every choice. We now provide (see our reply to one of your points above) an alternative choice rule which uses optimal (but non-uniform) fixed probabilities to choose between f,F,C,N ants. This fixed-probability rule, despite the degrees-of-freedom optimized for best match to data, still underperforms the sequential rule that operates on the dynamically updated load information. We understand (and hope) that this answers your query.

In the interest of full disclosure: our data and the corresponding analysis support (and favor) a decision rule that targets an ant for allogrooming in a way that is monotonically and stochastically related to its current load. This rule could be mechanistically implemented in a variety of ways. Assuming global knowledge of current spore loads, the ant could “flip a biased coin” and choose. Assuming only local knowledge and no comparisons, the ant could perform “sequential choice” and use a particular decision function. These variants are difficult to distinguish simply based on behavioral switch rate to allogrooming. There is correlative evidence (in addition to biological plausibility), however, to support the sequential model, which we now make more explicit, along with the caveats about what can or cannot be identified directly from data (main text, starting line 234).

Line 169: This sentence contrasts the sequential-choice rule with an alternative where ants were choosing targets “randomly”. Technically, both strategies are “random”, but one of them uses a probability to decide whether to accept or ignore an encountered ant, and the other chooses an ant from a uniform distribution. Avoid using “randomly” when what is probably meant is “uniformly.” We have changed throughout the main and supplementary text and figure to “uniform random”.

Line 200: “measurable benefit at the collective level” -- Compared to what? Do you have a no-behavior benchmark to compare to?

As detailed above, we have run an additional no-choice experiment (Fig. 7b), which, as we think, provides compelling evidence for a functional benefit. We rephrased to (lines 354-362): *Taken together, the observed higher spore removal of better choosers (Fig. 7a), as well as the observed higher spore removal of groups with choice (Fig. 7b), indicate that many small grooming biases in individual ant decisions accumulate to a functional benefit at the collective level. The ability of individual ants to preferentially target higher-load individuals (rather than choosing uniformly at random; Fig. 3d, Supplementary Model), combined with the higher grooming efficiency at higher spore loads (Type II functional response model, see above), builds up to highly efficient pathogen removal at the group level. This effect is already detectable for our experimental groups of only six ants, in line with previous reports that collective benefits can emerge already at these very small group sizes³⁶.*

Line 201: I had a difficult time reading the sentence coming from line 200 on the previous page. I would recommend removing both commas (before “overall” and after “group”).

We reformulated this section (as detailed above) and have simplified the sentence structure to (lines 337-340): *We quantified how grooming decisions affected a group’s spore removal efficiency, that is, the proportion of spores that the ants removed in the 90 min after exposure from the treated ants by collecting them into the nestmates’ infrabuccal pockets and inactivating them as pellets.”*

Line 204: Use of “random” again. The sequential-sampling model is ALSO randomly choosing. “Choosing uniformly” seems better than “choosing at random”.

As detailed above, we changed to “uniform random” throughout the main text and the supplement, as suggested.

Line 205: “builds up to highly efficient pathogen removal” -- relative to what benchmark? What is the control?

Here, we previously compared only the correlative evidence of ‘better choosers’ vs ‘worse choosers’ in terms of their total spore removal. In the revision, we have substantially expanded the evidence for the importance of choice, by performing an entirely new experiment where we compare ant groups in which the choice-to-groom is possible against cases where it is not. Our results strongly support the idea that the collective (total) spore removal will be significantly higher where the choice – in particular, informed choice to groom preferentially the higher-load individual – is possible. We explain the new setup (which comes with a new figure as well) in the revised

manuscript, and state specifically in lines 340-344: *“We found that groups which had more frequently successfully targeted the higher-load individual (‘better choosers’) removed more spores than groups which less frequently picked the higher-load ant (‘worse choosers’; Fig. 7a, Supplementary Fig. 10). We next evaluated how a complete prevention of choice would affect spore removal efficiency, by performing a separate ‘functional knock-out experiment. ...”*

Line 214: It is stated that the "maximum rule" requires "remembering pathogen loads on every group member". Certainly a sequential-sampling model with a plastic deterministic threshold would only have to remember the highest load it has seen so far. If that threshold started high and decayed over time, then eventually only the highest would be serviced, but the agent would not need to remember all loads at the same time.

Thank you for highlighting this. We have reformulated to (lines 270-272): *“It would require each ant to assemble global information by assessing and remembering the maximum pathogen load of its group members, to subsequently cognitively identify and physically locate the most infectious individual to groom.”*

Line 220: "convincingly" can be a little off-putting for readers, particularly those who are not yet convinced

We removed the word “convincingly”. The sentence now reads (lines 276-279): *“As soon as probing other ants incurs any time cost, our simulations (Supplementary Model) show that the experimentally-motivated sequential-choice rule (based on ‘cheap’ partial information) will outperform the hypothetical maximum rule (based on costly complete information) as colony size increases (Fig. 5),...”*

Line 223: Unclear that exploration--exploitation tradeoff is really the right tradeoff. Seems more like speed--accuracy tradeoff in this case.

See our detailed response above.

Line 223: Rather than considering a "maximum model" where the ant has to slow down to pay a time cost to take a pathogen measurement, it may also be worthwhile to consider an ant that can take instantaneous pathogen measurements so long as it is moving sufficiently slow. The classic work of Gendron and Staddon (1983) discussed how sensory constraints can limit the speed of predators, which reduces their intake rate but also allows them to actually find prey. Reducing speed is another kind of time cost. Perhaps ants stopping to antennate each other is more like the model you have in mind (but were such session observed?), but I just wanted to point out that there are other ways in which motion can accommodate higher sensing accuracy.

Thank you for this thought! While our work did not look at the statistics of ant motion in space (speed, positions etc.), we did provide further analyses both of data and of simulations to clarify the tradeoffs involved.

First, we “zoomed in” into two replicates of the data to see if we can find signatures of “sequential choice” that would imply transient probing events carried out by the focal ant before it commits to allogrooming. We now report on antennation events that could constitute such probing (paragraph starting in line 234) as well as provide a visualization of these events (Supplementary Figure 7). Second, we have further expanded the analysis of the “exploitation-exploration” tradeoff whereby a certain effective “time-cost” is associated with such probing, by analyzing what are the essential characteristics of the choice-to-groom probability based on the load that ensure fastest collective spore removal. We report on that in Supplementary information (pages 38-42) and additional supplementary figure (Supplementary Figure I), which we summarize in the main text as follows (lines 280-286): *“While the exact dependence of grooming probability on the observed load in the sequential-choice rule that leads to most efficient spore removal depends on various factors, all efficient rules share a very small probability of targeting a non-infectious ant for grooming (Supplementary Model), as we observe in the data. We note that even in small groups as used for our*

experiment, where ants could conceivably gather global information and use the optimal maximum rule without much efficiency cost (Fig. 5), our data preferentially support the sequential-choice rule.

Line 225: "convincingly" again

We removed "convincingly", see in last sentence of the paragraph in the preceding comment (lines 284-286 in the manuscript).

Lines 237--240: Here, it initially states that grooming suppression occurs in all ants even in the absence of pathogenic threat", but then it concludes that "untreated nestmates remained nearly unchanged." These two statements seem to be contradictions.

We have corrected this section (lines 311-314): "*Grooming suppression is a general response to received grooming – it occurs in all ants even in the absence of a pathogenic threat (i.e. in the pre-treatment period). The suppressive effect increased ~2-fold compared to the pathogen-free pre-treatment situation for the treated ants, while it was reduced ~2-fold for the untreated nestmates (Fig. 6b).*" We hope that this makes clear that the suppressive effect of grooming (which also occurs at baseline levels in the absence of pathogen in the pre-treatment period) is a general effect, which is subsequently differentially modulated for the treated ants vs. nestmates. See also our additional comment below.

Line 250: Another use of the word "random" when "uniform" is probably more accurate. Replaced as detailed above.

Line 287: Should "assure" be "ensure"? Or "help to ensure"?

We changed to (lines 392-395): "*The identified rules contribute to preferential grooming of the most-infectious colony members by the individuals with the lowest own infectiousness, identified by epidemiological modeling as important to reduce disease risk for the colony*³³."

Line 475, Figure 1e: I recommend adding color codes in a legend somewhere.

We added color information in the legend (new Fig. 2a): "*Red indicates spores originally applied to the F-individual, yellow to the f-individual.*"

Line 475, Figure 1g: Rather than using arbitrary bins just to plot this histogram, why not present the expected CDF and the empirical CDF? That will eliminate the need for binning and will make the Kolmogorov--Smirnov distance between distributions more visual.

We here present the CDFs:

While we agree that generally plotting CDFs is preferable since their empirical estimates can be constructed without binning the random variable, we feel that the comparison of CDFs in this case doesn't add to clarity (basically, because the sigmoid shape that all CDFs have by construction over-emphasizes the similarity between the observed distribution and null expectation, and does not highlight their difference). We would therefore prefer to keep our original plot (now Fig 2c), constructed based on binned estimates of empirical or bootstrapped null PDFs, and a visualization of their difference.

Line 494 and 495: "does not systematically depend on current load proportion (42/45 pairwise KS tests $p \leq 0.05$)" -- Is the $p \leq 0.05$ correct? If so, then is the "does not systematically depend" correct? Thank you very much for alerting us to this typo. We corrected it to " $p > 0.05$ ".

Line 503, Figure 2: As described above, would be good to plot performance of sequential sampling model with acceptance probability fixed (but optimized).
As detailed above, we included that alternative rule and consequently expanded the figure (new Fig. 3) accordingly.

Line 533, Figure 4d: Why is there an initial difference between nestmates and pre-treatment? Thank you very much for pointing this out. We had previously overlooked the details and biological relevance of this finding. The nestmates receive very little grooming in the post-treatment phase (less than in the pre-treatment phase) but they groom a lot more than in the pre-treatment phase due to the presence of spores. We understand the initial difference as follows: after receiving grooming the nestmates in the post-treatment phase are more likely to groom right away (smaller suppression). As these events are not only limited to the start of the experiment, we consider it likely that this could reflect their experience with pathogen load that they encountered. We now write (in lines 312-316): "*The suppressive effect increased ~2-fold compared to the pathogen-free pre-treatment situation for the treated ants, while it was reduced ~2-fold for the untreated nestmates (Fig. 6b). Untreated nestmates were thus less reactive to the suppressive effect of being groomed after pathogen exposure of their group members.*"

Line 636: "received a different VARIANT of the two distinctively..." (word "variant" is missing)
We changed to clarify: "*each ant received one of the two distinctively labelled spores, i.e. either the GFP- or RFP-variant of the same M. robertsii strain (randomized between replicates).*"

Line 679: The mechanism underlying how the untreated nestmates detected that they should reduce their poison uptake behavior is not obvious to me. It seems like it should be commented on. We now speculate that the reduced poison uptake behavior of the nestmates may occur in response to the release of poison by the treated individual, which they may detect due to its volatile nature (lines 143-147): "*It is still unclear whether nestmates refrain from using own poison when sensing the increased application of the volatile formic acid-rich disinfectant by the treated individuals, or whether they can assess their low risk of getting cross-contaminated with high, disease-causing pathogen levels⁴¹.*"

Supplementary Material, page 19: Extra space before "where y_i are the centers..."
Corrected.

Supplementary Material, page 23: "standard SSE algorithm" should probably be "standard SSA algorithm".
Corrected.

Supplementary Material, page 27: See comments above about how Figure E demonstrates that a fixed-probability sequential-sampling model would likely perform very well.

See above reply.

Supplementary Material, page 31: Probably more like a speed--accuracy tradeoff than an exploration--exploitation tradeoff.

See above reply.

Thank you for the opportunity to review this manuscript.

--Theodore (Ted) P. Pavlic

Thank you for your many suggestions! We feel that they have substantially improved the clarity and scientific content of our manuscript.

Reviewer #3 (Remarks to the Author):

Review of "Dynamic pathogen detection and social feedback shape collective hygiene in ants" by Barbara Casillas-Pérez et al. for Nature Communications

It is well known that social insects are collectively able to suppress the spread of diseases in their societies. Actually, this behavior is a requirement for their long-lived and prolific nests. Most research on the underlying mechanism has been conducted in honeybees and particularly in ants; the Cremer group is at the fore-front of this research. Here again they bring the research field to a new level, by combining individual observations on grooming and its effectiveness with simulations of individual decisions that could explain the collective disease management patterns in the studied ant species (and probably also other eusocial insects). Behavioral analyses revealed that ants in groups of six increase grooming and preferentially target highly-infectious conspecifics when perceiving high pathogen load, but reduce grooming after having been groomed by others. Individual decisions rely only on local and incomplete information, but simulations suggest that such decision rules can lead to efficient collective disease defense of whole nests.

Overall both the new methods and the simulations applied will bring the field a major step forward and will inspire many other researchers working with this and other model systems. The article is perfectly written, very well illustrated and obviously went through several rounds of expert review (also seen in the acknowledgements section). I find methods and results very well explained and introduced, and all claims and conclusions fully supported. The findings are very exciting and Nature Communications is a great fit for this important article.

Thank you very much for the appreciation of our work.

During reading the article I came up with several comments that were all finally resolved in the supplementary methods section. However, two of these things I suggest to briefly mention already in the main manuscript instead of the supplements: (1) obviously it is very important that observers are blind to the manipulations of individuals, so it is great that observers were blind to the color coding – please mention that in the main text already;

We have now prominently clarified this point in the main text: first, when introducing the experimental design (lines 102-104); and second, in the new method section (lines 466-470): "As ant treatment had been applied to two individuals that were randomly chosen independent of their color, no association could be made by the observer between ant color and treatment, neither within nor across replicates and treatment groups. Ant color-to-treatment assignment was also not revealed in the videos, ensuring bias-free behavioral scoring." We also expanded that this was not only the case for the behavioral observations, but also for further sample processing for spore quantification (line 494-495): "GFP- and RFP-usage was randomized across ant treatments, colors and replicates and sample labels did not contain treatment information."

(2) it is great that you quantified spore loads on individuals after applying your treatments, but this is not mentioned in the main text – please change that. Furthermore, I found Fig 1a misleading in this respect. This figure looks like a result of your study (it is presented together with other results), but is actually a description of the setup. However, you have quantified spore loads, so instead of using just dummy numbers, please use the actual numbers with variances you give in the supplements.

We added this information into the legend of Fig 1a, where these data (and indeed no dummy numbers) are plotted.

Additionally, we made further clarifications as follows. In line 152 and onwards we say: *“Together, individual and collective sanitary behaviors led to a >80% spore reduction on the exposed individuals from the initially-applied spore load.”* and we extended our experimental design description (lines 96-102) to clarify: *“We treated two of the workers with either a high (F) or a low (f) dose of the fungal pathogen Metarhizium robertsii, or a non-pathogenic control (C), and combined them to create groups differing both in overall pathogen load and the load differences between the two treated individuals. The untreated four nestmates therefore either faced a clear (FC, fC), less distinct (Ff) or no (FF, ff, CC) initial spore load difference between the two treated individuals (Fig. 1a; Supplementary Table 1; n=16-17 replicates per group treatment).”* We also added the relevant information in the methods section, which is now an integral part of the manuscript (lines 490-495).

Importantly, even though Fig. 1a is based on real data of the initial spore loads of both F and f individuals after dipping into the spore suspension, these values are not directly the values for the ants used in our experiment. This is because to measure these initial values, the ants had to be frozen immediately after dipping, whilst the spore load of the experimental ants could only be determined after the experiment. Thus, while the ants used to determine the initial load measurement and the ants used in the main experiment are not identical, they have undergone the same type of experimental handling and thus should be statistically indistinguishable. Furthermore, the sample sizes differed as we obtained the initial loads for 30 individuals each of F and f, while using 66F and 65f (in their respective combinations with other treated ants, Supplementary Table 1) in the experiment.

Lastly, adding the error bars would show the variation of the group-level spore load, whereas our main focus lies in the difference between the two individual treated ants, which would not be implied by such visualization. We do provide the relevant detail in the supplement (where we explicitly model spore dynamics based on the distribution of the initial spore loads).

Taken together, we would therefore prefer to keep our original Fig. 1a as is, having clarified the potential misunderstandings about whether these are only dummy numbers or not in the figure legend.

So apart from these two minor comments I have no further suggestions to change before acceptance for publication. However, I just would like to point out that I am an experimental entomologist and do not feel competent to evaluate the simulations part.

Reviewers' Comments:

Reviewer #1:

Remarks to the Author:

The manuscript has been improved and can be accepted for publication in Nature Communications now. I believe that this paper will be helpful for the entomological researchers in the social immunity and social behaviors.

Reviewer #2:

Remarks to the Author:

The authors have done a very laudable job responding to reviewer comments from their first submission. This revision has expanded their results in exciting new directions (beyond what I was expecting) and answers the concerns that I had. I am happy to recommend that the updated version of this manuscript be accepted for publication.

My only minor comments for the authors are as follows (which are responses to comments made in their rebuttal letter).

* The authors have chosen to stick with exploration--exploitation tradeoff language (in lieu of, for example, speed--accuracy tradeoff framing). Part of their justification is that they are familiar with foraging literature that frames similar problems in terms of exploration and exploitation. While I agree that this is a common framing, I think that it has often been misused and has diverged very far from the original context for which "exploration--exploitation" had the most utility. This is similar to (for example) saying that a Prisoner's Dilemma is a good model for a tragedy of the commons; whereas a Prisoner's Dilemma does model a generic cooperation tragedy, there is no commons problem (for which a Hawk--Dove model is a much better example of a problem related to negative externalities and common-pool resources). All of that being said, it is important to maximize clarity of communication to likely readers, and I agree that most readers who will likely read this paper will be less annoyed by the use of exploration and exploitation in this way. So, to answer the authors' question, I would not insist on them shifting to a different tradeoff. I'm fine with them sticking with the current tradeoff -- particularly if they do so with significant deliberation. In other words, so long as they themselves feel they are prioritizing accuracy over speed, then I'm fine with them describing their results in terms of exploitation and exploration.

* The authors have chosen to stick with their arbitrary binning in Figure 2c to compare empirical and expected distributions. They plotted the two CDF's together in the rebuttal and said that it is less clear than the histogram in Figure 2c. For me, I am still far less convinced by the histogram in Figure 2c (and more convinced by the comparison of CDF's in the rebuttal). I wonder if the authors considered using probability plots (such as a QQ plot and/or a PP plot) to more rigorously compare these distributions graphically. I only suggested plotting the CDF's as it maps nicely to the KS test. But, if a visualization is necessary regardless of whether it connects to the test, then I would think that probability plots (QQ alone or QQ and PP together (but probably not PP alone)) would work fine for this and be more conventional. That said, I have no desire to "die on this hill." It is my guess that most readers will be happy with the Figure 2c visualization as it is and will lean on the stats if they need more convincing.

So, while I offer up those two comments for the authors' consideration, I do not think that either of those comments should hold this manuscript up. The authors have done a great job answering my concerns (and the concerns of the other two reviewers, I think), and I think it will be exciting to have this manuscript join the archival record soon.

Reviewer #3:

Remarks to the Author:

Thanks for the revision of the manuscript. All my comments were addressed. I recommend the ms to be published as it is

We thank all three reviewers for their positive evaluation of our revised manuscript. Below we answer to the points raised by Reviewer 2.

REVIEWERS' COMMENTS

Reviewer #1

The manuscript has been improved and can be accepted for publication in Nature Communications now. I believe that this paper will be helpful for the entomological researchers in the social immunity and social behaviors.

Thank you.

Reviewer #2

The authors have done a very laudable job responding to reviewer comments from their first submission. This revision has expanded their results in exciting new directions (beyond what I was expecting) and answers the concerns that I had. I am happy to recommend that the updated version of this manuscript be accepted for publication.

Thank you.

My only minor comments for the authors are as follows (which are responses to comments made in their rebuttal letter).

* The authors have chosen to stick with exploration--exploitation tradeoff language (in lieu of, for example, speed--accuracy tradeoff framing). Part of their justification is that they are familiar with foraging literature that frames similar problems in terms of exploration and exploitation. While I agree that this is a common framing, I think that it has often been misused and has diverged very far from the original context for which "exploration--exploitation" had the most utility. This is similar to (for example) saying that a Prisoner's Dilemma is a good model for a tragedy of the commons; whereas a Prisoner's Dilemma does model a generic cooperation tragedy, there is no commons problem (for which a Hawk--Dove model is a much better example of a problem related to negative externalities and common-pool resources). All of that being said, it is important to maximize clarity of communication to likely readers, and I agree that most readers who will likely read this paper will be less annoyed by the use of exploration and exploitation in this way. So, to answer the authors' question, I would not insist on them shifting to a different tradeoff. I'm fine with them sticking with the current tradeoff -- particularly if they do so with significant deliberation. In other words, so long as they themselves feel they are prioritizing accuracy over speed, then I'm fine with them describing their results in terms of exploitation and exploration.

Thank you very much for these additional details. We have now included an additional section as well as references in the main text, which we hope will account for this. It reads: *"As soon as probing other ants incurs any time cost, our simulations (Supplementary Note 2), show that the experimentally-motivated sequential-choice rule (based on cheap partial information) will outperform the hypothetical maximum rule (based on costly complete information) as colony size increases (Fig. 5), in a classic manifestation of the exploration-exploitation tradeoff^{48,49}. Here, exploration refers to the effort by nestmates to locate the ant with highest spore load, where grooming (exploitation) would be most efficient. This is in analogy to the application of the same tradeoff to the problem faced by foraging animals that*

need to balance the time spent on searching for new and possibly rich foraging grounds, with the time spent exploiting known, but perhaps more meagre, grounds. We point out that the decisions faced by the nestmates could also be understood in terms of the speed-accuracy trade-off⁵⁰⁻⁵². Here, nestmates need to invest more time (i.e., to probe colony members) to make a more accurate choice (i.e., to locate the highest-loaded individual and deliver most efficient care.”

* The authors have chosen to stick with their arbitrary binning in Figure 2c to compare empirical and expected distributions. They plotted the two CDF's together in the rebuttal and said that it is less clear than the histogram in Figure 2c. For me, I am still far less convinced by the histogram in Figure 2c (and more convinced by the comparison of CDF's in the rebuttal). I wonder if the authors considered using probability plots (such as a QQ plot and/or a PP plot) to more rigorously compare these distributions graphically. I only suggested plotting the CDF's as it maps nicely to the KS test. But, if a visualization is necessary regardless of whether it connects to the test, then I would think that probability plots (QQ alone or QQ and PP together (but probably not PP alone)) would work fine for this and be more conventional. That said, I have no desire to "die on this hill." It is my guess that most readers will be happy with the Figure 2c visualization as it is and will lean on the stats if they need more convincing.

We have added a new Supplementary Fig. 6 that shows both the CDFs and the QQ plot.

So, while I offer up those two comments for the authors' consideration, I do not think that either of those comments should hold this manuscript up. The authors have done a great job answering my concerns (and the concerns of the other two reviewers, I think), and I think it will be exciting to have this manuscript join the archival record soon.

As detailed above, we integrated the speed-accuracy tradeoff in the main text and the CDF and QQ plot as an additional supplemental figure.

Reviewer #3

Thanks for the revision of the manuscript. All my comments were addressed. I recommend the ms to be published as it is

Thank you.